# Reconstructing the electrical structure of dust storms from locally observed electric field data

Huan Zhang [1,2] & You-He Zhou [1,2 ✉]

While the electrification of dust storms is known to substantially affect the lifting and transport of dust particles, the electrical structure of dust storms and its underlying charge separation mechanisms are largely unclear. Here we present an inversion method, which is based on the Tikhonov regularization for inverting the electric field data collected in a near-ground observation array, to reconstruct the space-charge density and electric field in dust storms. After verifying the stability, robustness, and accuracy of the inversion procedure, we find that the reconstructed space-charge density exhibits a universal three-dimensional mosaic pattern of oppositely charged regions, probably due to the charge separation by turbulence. Furthermore, there are significant linear relationships between the reconstructed space-charge densities and measured $PM_{10}$ dust concentrations at each measurement point, suggesting a multi-point large-scale charge equilibrium phenomenon in dust storms. These findings refine our understanding of charge separation mechanisms and particle transport in dust storms.

[1] Key Laboratory of Mechanics on Disaster and Environment in Western China Attached to The Ministry of Education of China, Lanzhou University, Lanzhou 730000 Gansu, P.R. China. [2] Department of Mechanics and Engineering Science, College of Civil Engineering and Mechanics, Lanzhou University, Lanzhou 730000 Gansu, P.R. China. ✉email: zhouyh@lzu.edu.cn

D isperse two-phase flows, a huge number of discrete particles or droplets embedded in the turbulent flows, are widespread in nature, industry, and even on other planets[1–8]. There are many examples of interest in disperse two-phase flows, such as sand saltation[9–14], dust devils[15,16], dust and sand storms[17–27], blowing snows[28,29], thunderstorms[30], volcanic eruptions[31–33], fluidization beds[8,34], as well as dusty plasmas[2,35]. In these systems, very intense electric field (E-field) and even lightning have been frequently observed due to particle electrification. The fact that electrification plays a key role in the lifting and transport of dust particles has been recognized by the scientific community. To date, considerable efforts have been put forth to explore the particle-static interactions in various conditions[1–8]. For example, during dust events, electrostatic forces could facilitate the lifting of dust particles from the ground by a factor of ten[25] and even directly lift sand particles from the surface if the ambient E-field up to 300 kV m$^{-1}$ can be reached[36]. The propagation of electromagnetic waves in dust storms was also found to be dramatically affected by the airborne charged dust particles[37]. In volcanic plumes, electrostatic forces may contribute to the formation of particle aggregation, thus affecting the dispersal and deposition of volcanic ash[38]. In fluidized beds, particle electrification could cause particles to adhere to the walls, thereby inhibiting particle transport[8,34]. In dilute granular flows, charged particles could be trapped in their mutual electrostatic energy well and thus form clusters[39]. Furthermore, electrostatic forces may also be an important factor in the aggregation of cosmic dust and the formation of planetesimals[2,35]. Thus, owing to its great importance, a detailed understanding of particle electrification in granular systems is necessary[1–8].

Dust storms are highly complex polydisperse particle-laden turbulent flows with a very high Reynolds-number (typically of ~10$^7$ or greater)[40,41]. Although reports of particle electrification in dust storms could date back to Rudge's research in 1913[17], such electrification processes are still largely unclear[1,3–7]. The most obvious difficulty is that little information is available on the structures of the space-charge and E-field in dust storms, particularly at higher altitudes, due to the limitation of the measurement techniques and complexity of the structures themselves[1,5,7].

From the limited E-field measurements, previous studies inferred that the charge structure of dust storms was either monopolar or bipolar, but recent measurements suggested that the actual charge structure was probably more complex than previously recognized. The pioneering E-field measurements in dust storms by Rudge[17] found that the vertical component of the E-field in the near-ground region was directed upward, indicating that the finer dust particles at higher altitudes are negatively charged while the coarser sand particles near the ground are positively charged[3,42]. This simple model, which represents a downward-directed dipole moment, forms the preliminary assumptions of the charge structure in dust storms and is commonly referred to as a negative-over-positive structure[42]. Later measurements at the heights of about 1–2 m found a downward-pointing[25] or even alternating vertical E-field component[18,19] that continually reverses direction during dust storms. Williams et al.[19] further inferred that the charge structure was monopolar if the charge transfer between the ground and airborne dust particles was predominant, while the charge structure was bipolar if the charge transfer between airborne dust particles was predominant. It is worth noting that such monopole and bipolar charge structures have also been inferred in volcanic plumes based on three-dimensional (3D) lightning data[32,33]. However, recent 3D E-field measurements in dust storms using an atmospheric surface layer observation array have questioned these simple charge structures[26,27]. The direction of each component of the 3D E-field was found to vary with spatial location[27], which

cannot be explained by the monopole or dipole structure. Due to different responses to turbulent fluctuations, in fact, the oppositely charged particles with different sizes could be separated by turbulent eddies[43–45]. In this case, more complicated electrical structures could arise in dust storms, which motivates us to develop a method for determining the structures of space-charge and E-field in dust storms, both qualitatively and quantitatively.

According to Coulomb's law, we know that E-field at each point depends on the entire space-charge distribution in dust storms[46], which provides us a possible way to estimate the space-charge densities in a relatively large spatial extent based on the locally measured E-field data. To this end, we present an inversion method for inverting the E-field data measured in a near-ground measurement array to reconstruct the structures of space-charge and the E-field in dust storms. In mathematics, directly solving the space-charge and E-field is a typical ill-posed problem since the solution is nonunique and the solution procedure is unstable[47,48]. To solve this issue, the inversion method presented here is based on Tikhonov regularization[47,49], which is one of the pioneer methods of solving ill-posed problems.

The reconstructed electrical structures can be used to quantify the essential properties of dust storms. For example, previous studies[27] found that, at given ambient temperature (T) and relative humidity (RH), there were significant linear relationships between the dust concentrations and space-charge densities over the timescales of 10 min, suggesting a constant charge-to-mass ratio of dust particles (termed large-scale charge equilibrium phenomenon). This phenomenon has been previously verified at only one measurement point[25,27] and can be examined at multiple points based on the inversion results.

In this study, by performing a set of subsampling inversions, we demonstrate that the proposed inversion procedure is shown to converge as the subsampling size increases. The verification analysis shows that the residual between the normalized observed data and the model prediction is as low as 0.04 and the reconstructed space-charge densities agree excellent with the Gauss's law approximation (GLA)-based densities. Furthermore, we find that the charge structure of dust storms exhibit a universal mosaic pattern, where there are alternating charged regions of positive and negative polarities. Based on the estimated dust particle's Stokes number, we infer that such a mosaic charge pattern is attributed to the turbulence-driven separation of the oppositely charged dust particles. Finally, the large-scale charge equilibrium effects at multiple points are verified by the significant linear relationships between the reconstructed charge densities and the measured PM$_{10}$ (smaller than 10 μm in diameter) dust concentrations, from which the charge-to-mass (PM$_{10}$) ratio can be evaluated.

## Results
**E-field measurements during dust storms.** E-field data were collected at the Qingtu Lake Observation Array (QLOA), Gansu, China (Fig. 1a) between March 21 and June 2, 2017. Qingtu Lake is currently a large dry lake whose flat-lakebed covers nearly 20 km$^2$ (Fig. 1a). The QLOA site is situated between the Tengger Desert and the Badain Juran Desert and is frequently subjected to dust storms from March to May because of the Mongolian cyclones[50]. The prevailing wind direction in the QLOA site is northwesterly, suggesting that the main dust source area of the observed dust events is the Badain Juran Desert (Fig. 1a). The QLOA consists of one main observation tower (33 m in height) and over 21 auxiliary observation towers (5 m in height) arranged in a T shaped formation, thereby allowing us to perform the multi-point measurements of E-fields, dust concentrations, wind velocities, etc. A total of 19 vibrating-reed electric field mills

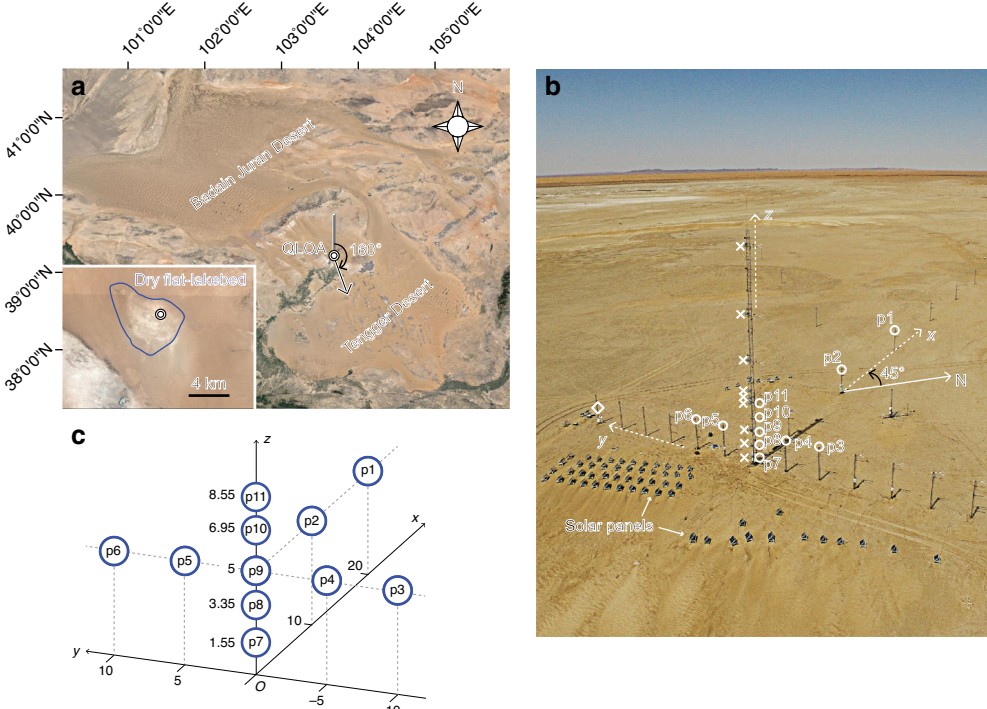

**Fig. 1 Location of the QLOA site and configuration of measurement array in this study. a** The QLOA site located at approximately 39°12′27″N, 103°40′03″E, and altitude around 1302 m. An arrow represents the prevailing wind direction at the QLOA site, where wind direction is measured in degrees clockwise from due north. The inset shows the area of the dry flat-lakebed of Qingtu Lake (credit: Google Earth Pro). **b** Cartesian coordinate system defined in this study. The positive direction of the *x*-axis points toward the northwest (at an angle of 45° anticlockwise from due north). The E-fields were measured at the points that are marked by open circles(∘) and labeled at p1-p11, the PM₁₀ dust concentrations were measured at the points that are marked by crosses (×), as well as the ambient T, RH, and visibility were measured at the point that is marked by a diamond (◇). The wind velocity was measured at p9. **c** The coordinates of the E-field measurement points p1-p11 are shown in units of meter.

(VREFMs) were deployed at the QLOA to measure the 3D E-fields within 30 m height above the ground during dust storms (Fig. 1b). The detailed arrangement of VREFMs is shown in Fig. 1c and Supplementary Table 1. The VREFM functions as a vibrating capacitor, where the ambient E-field component normal to the electrode is measured by detecting the induced charge on the electrode. The VREFM recorded data at 1 Hz with an uncertainty of approximately ±2.24% (see Supplementary Note 1 and Supplementary Figs. 1–3 for a brief description of the VREFM). In addition, a visibility sensor (Model 6000, Belfort80 instrument) was installed 1 m above the ground to measure visibility from 5 to 10,000 m with an accuracy of ±10% and a sampling frequency of 1 Hz. An ambient T & RH sensor (Model 41382, R. M. Young Company) was used to monitor ambient T and RH that are the major factors affecting particle electrification[25–27]. Nine DustTrak II Aerosol Monitors (Model 8530EP, TSI Incorporated) were installed at heights ranging from 0.9 to 30 m (Fig. 1b and Supplementary Figs. 4–6) to measure the PM₁₀ dust concentrations, with a sampling frequency of 1 Hz[26,27]. The 3D wind velocity at point p9 was measured by a sonic anemometer (CSAT3B, Campbell Scientific) at a sampled rate of 50 Hz. To identify the dust source areas, two dust collectors were mounted on the main tower near point p9 to collect the airborne dust particles during dust storms (Supplementary Fig. 7a).

During the 2017 field observations, over ten dust storms occurred and were fully recorded, but only three dust storms were used for inversion as we had obtained high-quality E-field data for them. In these storms, the maximum values of the streamwise wind speed, PM₁₀ concentration, and E-field intensity at 5 m height were ~15 m s⁻¹, 7.72 mg m⁻³ (corresponding to the visibility of ~90 m), and 180 kV m⁻¹, respectively (Fig. 2), which

suggests that these dust storms were very strong and were highly electrified. The wind directions of the three storms lay within 152.3 ± 4.7°, 160.9 ± 6.4°, and 171.4 ± 7.2° (in degrees clockwise from due north), respectively (Figs. 2c, f, and i), showing that all storms mostly originated from the Badain Juran Desert. The same dust source area for the three dust storms was also verified by the very similar size distributions and mineralogical compositions of the dust samples collected at point p9 (Supplementary Fig. 7b–7d).

The intensity evolution of these three dust storms behaved quite differently. The evolution of storm #1 can be qualitatively divided into three distinct stages: the first stage was termed the growth or developing stage, where the storm intensity (such as PM₁₀ concentration and E-field intensity) increased gradually with time; the second stage was termed the mature stage, where the storm generally reached a dynamic equilibrium state and its intensity remained at a relatively constant value during a period; the third stage was termed decay or dissipating stage, where the storm intensity decreased with time until the storm vanished (Figs. 2a and b). In contrast, no clear (or distinguishable long-period) mature stage was observed in storms #2 and #3. During storm #3, the storm intensity increased to its maximum value within 1.5 h then decreased without an obvious mature stage (Fig. 2g and h), while the storm intensity was repeatedly increased and decreased during storm #2 (Fig. 2d and e). In other words, there were several comparable peak intensities in storms #2, but only one peak intensity in storms #1 and #3.

**Constrained optimization for inverse space-charge**. The primary goal of the inversion is to reveal the 3D pattern of the space-charge density in dust storms, on the basis of the given E-field

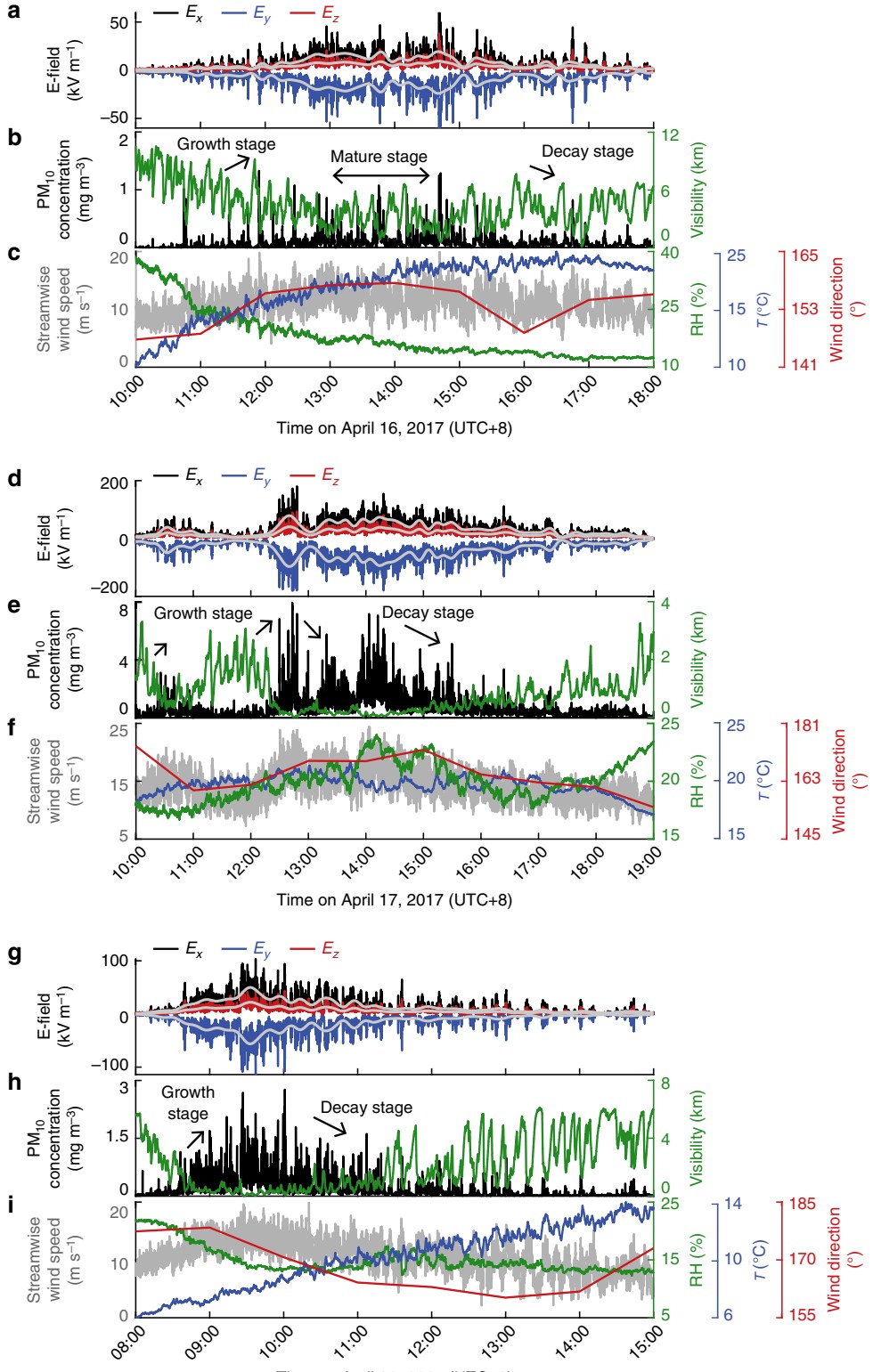

**Fig. 2 Examples of the measured data for the three observed dust storms. a–c** Time series of the 3D E-field, $PM_{10}$ concentration and visibility, as well as streamwise wind speed, wind direction, ambient temperature ($T$), and relative humidity (RH) data for storm #1. Here, E-field components $E_x$, $E_y$, and $E_z$ point in the positive direction of the $x$-, $y$-, and $z$-axis, respectively. As an example, the data of the 3D E-field, $PM_{10}$ concentration, streamwise wind speed, and wind direction are only shown at measurement point p9. **d–f** As in **a–c** but showing the data for storm #2. **g–i** As in **a–c** but showing the data for storm #3. Storms #1–#3 occurred on April 16, 17, and 20, 2017 (UTC+8), respectively. Gray lines in **a**, **d**, and **g** denote the time-varying mean of the E-field over the scales of $2^9$ s, which is extracted by the discrete wavelet transform method (see Methods).

measurements data. Theoretically, the forward problem that predicts E-field at a given point in dust storms can be formulated as an integration based on Coulomb's law and method of images if the space-charge densities are known[46], such that

$$\mathbf{E}(x, y, z) = \int\int\int_{\Omega} \mathbf{K}(\mathbf{r}, \widetilde{\mathbf{r}})\rho(\boldsymbol{v})\mathrm{d}^3\boldsymbol{v}, \quad (1)$$

in which the integral kernel $\mathbf{K}(\mathbf{r}, \widetilde{\mathbf{r}})$ is defined by

$$\mathbf{K}(\mathbf{r}, \widetilde{\mathbf{r}}) = \frac{1}{4\pi\epsilon_0}\left(\frac{\mathbf{r}}{|\mathbf{r}|^3} - \frac{\epsilon_r - 1}{\epsilon_r + 1}\frac{\widetilde{\mathbf{r}}}{|\widetilde{\mathbf{r}}|^3}\right), \quad (2)$$

where the computational domain is taken as $\Omega = [-L_x, L_x] \times [-L_y, L_y] \times [0, L_z]$, $\rho$ is the space-charge density per unit volume at source point $\boldsymbol{v} = (x', y', z')$, $\mathbf{r} = (x - x', y - y', z - z')$ and $\widetilde{\mathbf{r}} = (x - x', y - y', z + z')$ are the vectors pointing from the source point $(x', y', z')$ and the imaginary image charge point $(x', y', -z')$ to the field point $(x, y, z)$, respectively[46], $\epsilon_0 = 8.85 \times 10^{-12}$ C$^2$ N$^{-1}$m$^{-2}$ is the permittivity constant of air, and $\epsilon_r$ is the relative dielectric constant of the sandy ground. According to ref. [51], $\epsilon_r$ can be approximately taken as 5 in our model because of the sandy ground of the dry Qingtu Lake with low water content. The first term in the bracket of Eq. (2) accounts for the airborne charged dust particles and the second term accounts for the dielectric sandy ground, as the E-field in dust storms can be reasonably modeled as the charged dust particles above a planar dielectric sandy ground.

Given the E-fields data measured at the measurement points, the inverse problem of estimating the space-charge density based on Eqs. (1) and (2) is a typical Fredholm integral equation of the first kind, which can be discretized using the Galerkin method[48,52] (see Methods), and therefore rewritten in the matrix form

$$\mathbf{E}^{\mathrm{obs}} = \mathbf{G}\boldsymbol{\rho}, \quad (3)$$

where $\mathbf{E}^{\mathrm{obs}} \in \mathbb{R}^{m \times 1}$ is the E-field data vector measured at the measurement points, $\mathbf{G} \in \mathbb{R}^{m \times n}$ is the matrix form of the discretized forward model in Eq. (1), and $\boldsymbol{\rho} \in \mathbb{R}^{n \times 1}$ is the vector consisting of the unknown space-charge density.

In general, the discrete inverse problem, Eq. (3), is ill-posed and unstable with many solutions because collected data are finite in number and have unavoidable noise[48,52–54]. To obtain a single and stable model, we incorporate a priori information stating that the 2-norm of the solution is small so that the inverse problem can be formulated as a constrained optimization problem[47,48,52]

$$\min \quad \phi(\boldsymbol{\rho}) = \|\mathbf{E}^{\mathrm{obs}} - \mathbf{G}\boldsymbol{\rho}\|_{\mathbb{L}_2}^2 + \lambda^2 \|\boldsymbol{\rho}\|_{\mathbb{L}_2}^2, \quad (4)$$

where $\phi(\boldsymbol{\rho})$ is the most commonly used objective function known as the Tikhonov functional, and $\lambda$ is the regularization parameter. The first term on the right-hand side of Eq. (4) is referred to as misfit, which is a measure of the difference between the observed and the predicted data, and the second term is referred to as regularization, which constructs an evaluation of the agreement between the solution and the priori information. The regularization parameter $\lambda$ weights the contributions of the misfit and regularization terms to the minimization of the Tikhonov functional. Although a small $\lambda$ results in a solution that fits the observed data well, such almost no regularizations may always lead to an unstable and incorrect solution. In contrast, a large $\lambda$ leads to a solution that is closer to the priori information, thereby producing a large residual[53,54].

In brief, the proposed method should include: (i) obtaining E-field data from field observations; (ii) constructing a constrained optimization problem, Eq. (4), to estimate $\boldsymbol{\rho}$; and (iii) solving Eq. (4) using an optimal $\lambda$ (see Methods).

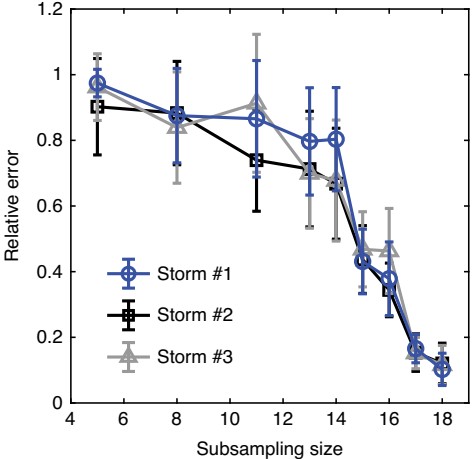

**Fig. 3 Relative error as a function of the subsampling size.** Relative errors are defined as $\|\boldsymbol{\rho}_{\mathrm{inv}}^m - \boldsymbol{\rho}_{\mathrm{inv}}^{19}\|_{\mathbb{L}_2}^2 / \|\boldsymbol{\rho}_{\mathrm{inv}}^{19}\|_{\mathbb{L}_2}^2$, where the subsampling size $m$ varies from 5 to 18. Here, the inversions for storms #1-#3 are performed at 13:20:00 on April 16, 2017 (UTC+8), 14:10:00 on April 17, 2017 (UTC+8), and 09:35:00 on April 20, 2017 (UTC+8), respectively. Error bars denote standard deviation over ten trials.

Notably, inversion performance is very sensitive to E-field fluctuations. In practice, the small-scale (high-frequency) fluctuations of the observed E-fields at a measurement point are dominated by turbulence and are probably due to the local changes in space-charge densities. Such small-scale and local changes at a point cannot be reflected at other points far from it, and thus could result in a failure of the inversion when the raw data are used. As shown in Supplementary Figs. 8–10, the small (locally large) E-field fluctuation leads to low (high) inversion residuals $\zeta$ (see Methods). In the following sections, the inversions are thus performed using the time-varying mean of the E-field series over the $2^9$ s timescales (see Methods), which is on the order of the integral timescale of the turbulence in the atmospheric surface layer[40].

**Verification of the inversion method.** To test whether the inversion converges as the subsampling size increases, we first perform the subsampling (random subset) inversion. The subsample data set $\mathbf{E}_m^{\mathrm{obs}}$ with subsampling size $m < 19$ is randomly selected from the total 19 measurement points. As in refs. [55,56], we execute each subsampling inversion ten times. Then, the reconstructed space-charge density and the relative error with respect to the original 19-point inversion were computed and averaged over the ten trials at each subsampling inversion. Figure 3 and Supplementary Figs. 11–13 illustrate how the subsampling inversion converges for the three dust storms. It can be seen that the relative errors decrease rapidly with increasing $m$ and are reduced to ~0.1 (or 10%) for the three dust storms (Fig. 3). In addition, there are almost no significant differences in charge patterns when the subsampling size exceeds 17 for each dust storm (Supplementary Figs. 11–13). This suggests that the densities $\boldsymbol{\rho}_{\mathrm{inv}}$ reconstructed from the complete 19-point measurement data are reasonable and reliable, where all relative errors are within 10%.

To examine the inversion accuracy, we then perform the residual analysis (see Methods) of the inversion using the complete 19-point measurement data. Overall, the inversion residuals $\zeta$ for storms #1-#3 are in the range of $0.04 \pm 0.003$ (Fig. 4a–c), suggesting that the predicted E-fields agree well with their measurements (Fig. 4d–f, Pearson's correlation coefficients $r > 0.99$). Meanwhile, the residuals are highly sensitive to data

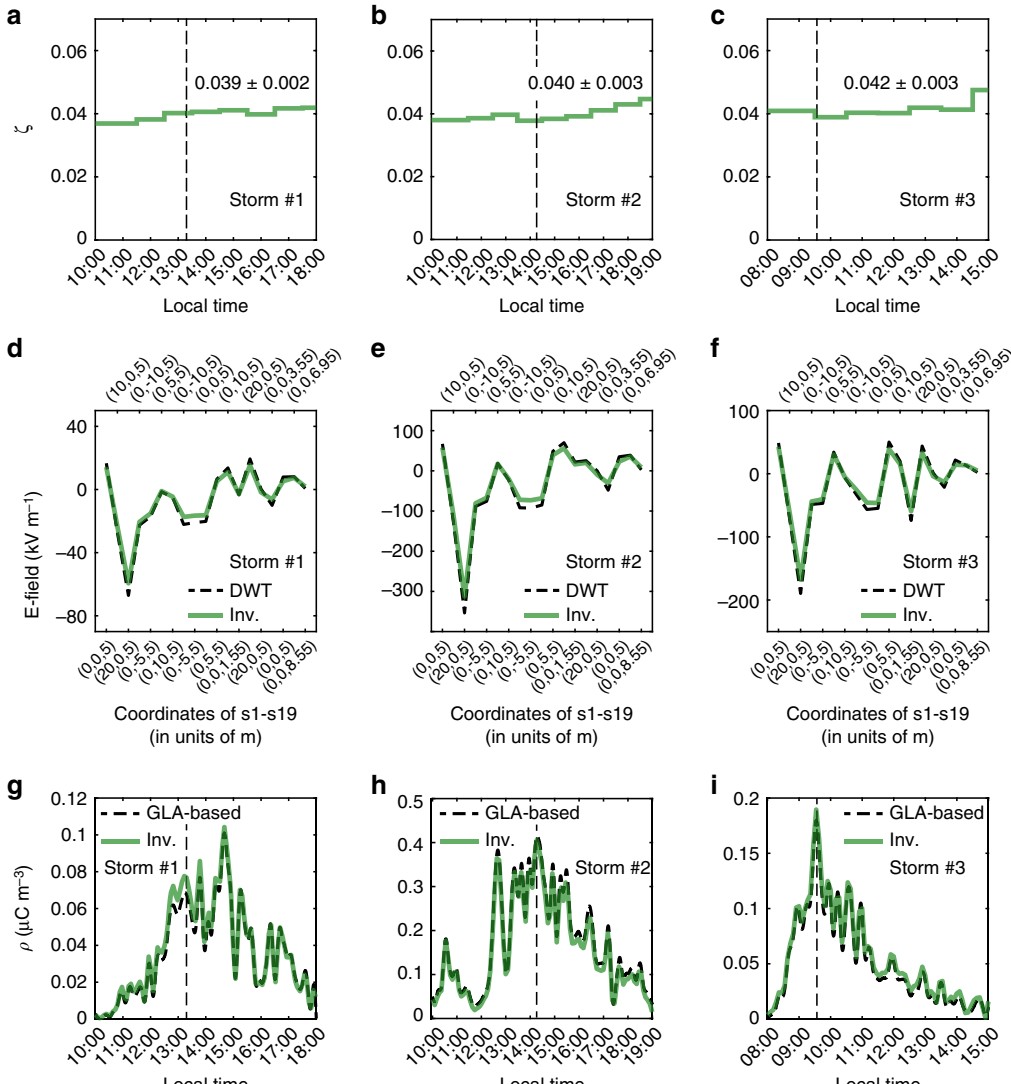

**Fig. 4 Verification of the inversion results. a–c** Residual distributions of the inversions for the three observed storms, where the normalized residual $\zeta$ are shown by the stairstep lines. **d–f** Comparisons of the E-fields predicted by the inversion model (Inv.) with the discrete wavelet transform (DWT) mean data of sensors s1-s19 at the three instants depicted by the dashed vertical lines in **a–c** and **g–i**. The coordinates of the sensors s1-s19 are shown at the top and bottom x-axis. **g–i** Comparisons of the reconstructed densities $\rho_{inv}$ with the Gauss's law approximation (GLA)-based densities $\rho_{GLA}$ at measurement point p9.

quality. Large residuals correlate with larger data disturbances and higher noise. For each storm, the residuals increase slightly with time, indicating that the long-period ambient noise and the instrument drift are negligible during measurements.

To further verify our inversion method, we compare the reconstructed densities $\rho_{inv}$ with the GLA-based densities[27] $\rho_{GLA}$ at point p9 (see Methods). As shown in Fig. 4g–i, the reconstructed densities $\rho_{inv}$ are in excellent agreement with $\rho_{GLA}$ ($r > 0.99$ and relative error $\| \rho_{inv} - \rho_{GLA} \|_{\mathbb{L}_2}^2 / \| \rho_{GLA} \|_{\mathbb{L}_2}^2 \sim 0.007\text{-}0.01$) during the three dust storms. Here, the maximum reconstructed densities $\rho_{inv}$ at point p9 is on the order of ~0.4 μC m⁻³, which is consistent with the measurements values of ~0.01–0.1 μC m⁻³ in dust storms by Kamra[18] and dust devils by Crozier[15] at ~1 m height above the ground, but is larger than the measurements values of ~5–25 pC m⁻³ in Saharan dust layer by Nicoll et al.[20] at altitude up to 4 km.

**Structures of space-charge and E-field.** Figure 5 and Supplementary Figs. 14–15 show the evolution of 3D structures of the space-charge density during storms #1 to #3, respectively. We

find that the reconstructed space-charge patterns exhibit a mosaic of positively and negatively charged regions. Such mosaic patterns consistently appeared in the whole duration of all three observed storms, suggesting that the mosaic charge pattern is a general feature of dust storms. Since the inversions are performed with the time-varying mean data, such reconstructed charge patterns are in fact an average pattern over the 2⁹ s timescales. In these cases, the instantaneous or small-scale changes in space-charge structure cannot be revealed, thus showing a very similar charge structure at different time points of each storm (e.g., a similar shape of isosurfaces at different times in Fig. 5). Also, the reconstructed charge structures of storms #1–#3 were almost identical because the meteorological conditions (e.g., mean wind speed and wind direction, Fig. 2) and dust source areas were almost the same (Supplementary Fig. 7). Importantly, the mosaic patterns are quite distinct from the previously inferred monopolar and bipolar charge structures in dust storms. Such mosaic patterns may be formed by the separation of oppositely charged particles by turbulence[44,45], as explained in the volcanic eruptions[31,33].

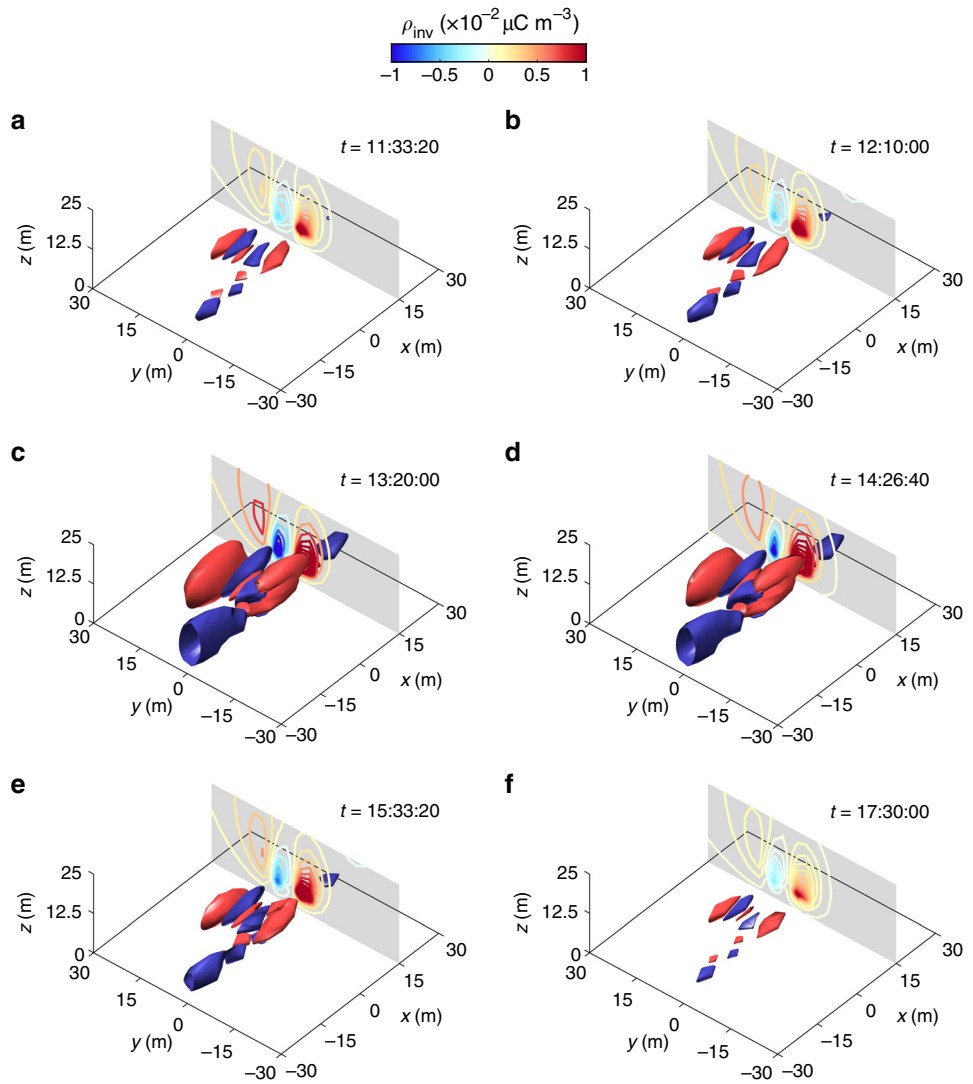

$\rho_{\mathrm{inv}}$ ($\times 10^{-2}\,\mu C\,m^{-3}$)

**Fig. 5 Evolution of the 3D structure of the space-charge densities during storm #1. a–f** The reconstructed space-charge densities $\rho_{\mathrm{inv}}$ at different stages of the observed dust storm. The isosurfaces are shown at a space-charge density magnitude of $2\times 10^{-2}\,\mu C\,m^{-3}$; the positive surfaces are colored in red, while the negative surfaces are colored in blue. Times $t$ are shown as the local time on April 16, 2017 (UTC+8). Contourslices at $x = 15\,m$ are colored based on the space-charge densities.

Based on the reconstructed densities $\rho_{\mathrm{inv}}$, the 3D E-field in the computational domain can be predicted by the forward model [i.e., Eqs. (1) and (2)], as shown in Fig. 6 and Supplementary Figs. 16–17. Since the mosaic structure of space-charge density consistently existed in the observed dust storms, the reconstructed E-fields were not uniformly oriented. This orientation change suggests that the E-field in dust storms is a 3D field[27] that distinctly differs from the one-dimension E-field model in pure sand saltation[9–14].

**Multi-point large-scale charge equilibrium.** The reconstruction of 3D space-charge densities in dust storms allows us to evaluate the ratio of densities $\rho_{\mathrm{inv}}$ to $PM_{10}$ concentration [termed charge-to-mass ($PM_{10}$) ratio hereafter] at each measurement point, which is similar but not equal to the actual charge-to-mass ratio because in addition to $PM_{10}$, charged particles larger than $10\,\mu m$ have also contributed to the space-charge densities. In such a case, the charge-to-mass ($PM_{10}$) ratio is larger than the actual charge-to-mass ratio of dust particles. As shown in Fig. 7 and Supplementary Figs. 18–19, at given ambient $T$ and RH, the

reconstructed space-charge densities are linearly related to the mean $PM_{10}$ concentrations ($R^2 \sim 0.5$–$0.9$, $p$ value $< 0.0001$). Thus, the charge-to-mass ($PM_{10}$) ratio at each measurement point of $PM_{10}$ can be evaluated by the slopes of the linear-fit lines in Fig. 7 and Supplementary Figs. 18–19. A constant charge-to-mass ($PM_{10}$) ratio at each point suggests that, on average (over the scales of $2^9$ s), the dust particles have acquired a dynamic charge equilibrium passing through each measured point, which was previously reported but only at one height[25–27,57,58].

Although the reconstructed space-charge and E-field structures seem very similar among the three dust storms, the vertical profile of the charge-to-mass ($PM_{10}$) ratio varies from storm to storm. As shown in Fig. 8, the strongest charge-to-mass ($PM_{10}$) ratio of particle charging occurred during storm #3 while the weakest case occurred during storm #2. The different levels of electrification for storms #1–#3 were likely to be caused by the remarkable changes in ambient $T$ and RH (Fig. 2)[25–27]. On the other hand, the vertical profiles of the charge-to-mass ($PM_{10}$) ratio are not identical among the three dust storms. For example, at 8.5 m height, the charge-to-mass ($PM_{10}$) ratio was negative during

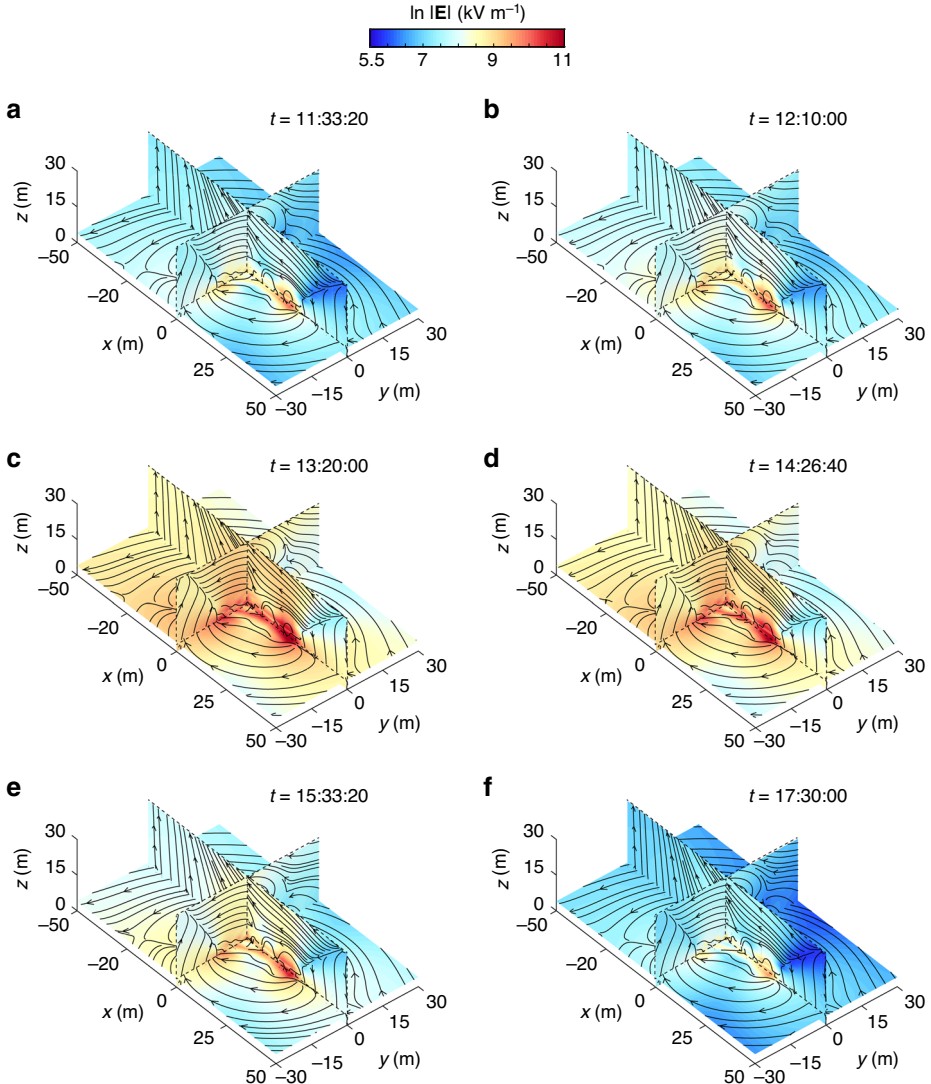

**Fig. 6 Evolution of the 3D structure of the E-fields during storm #1. a–f** E-field predicted from the reconstructed space-charge densities at different stages of the observed storm. Slices at $x = 0$ m, $y = 0$ m, and $z = 4$ m are colored based on the log-magnitude of the 3D E-field, $\ln |\mathbf{E}|$. Times $t$ are shown as the local time on April 16, 2017 (UTC+8). Lines represent the E-field lines.

storms #1 and #2, but was positive during storm #3. Such a difference may be caused by the slight change in the particle size distribution of storm #3 (Supplementary Fig. 7b and c).

## Discussion

To resolve the structures of space-charge density and E-field in dust storms, we introduce a constraint that requires the reconstructed (or regularized) solution with a small 2-norm which is the standard form of the Tikhonov regularization[47]. The sub-sampling convergence and residual analyses are performed to verify the accuracy of the inversion procedure. The proposed inversion method exhibits a good fit to the observed data and is in excellent agreement with the GLA-based measurement results[27]. Since the spatial resolution of the inversion is relatively low, at $5 \times 2.5$ m$^2$ in the horizontal plane and 0.1–27.7 m in the $z$-direction (see Methods), the inversion model cannot precisely resolve smaller-scale structures. The limitations regarding the inversion resolution mainly arise from the large VREFM sensor spacing, which was ~10 m, 5 m, and 1.75 m in the $x$-, $y$-, and $z$-direction, respectively. In future work, the inversions could be improved by incorporating additional priori information

associated with the solutions and more measurement data collected with a larger spatial extent and smaller sensor spacing.

In this study, we reveal the 3D mosaic charge structure of dust storms, which is physically more reasonable. According to the directions of the measured E-field, researchers inferred that the charge structure of dust storms was monopolar or bipolar[19]. Interestingly, the monopolar and bipolar charge structures were also inferred in volcanic plumes based on the 3D lightning data[32,33]. However, a more refined structure cannot be inferred by previous studies. In dust storms, existing E-field measurements were only performed in a very narrow region near the ground[5,7]. In volcanic plumes, the lightning-based method cannot incorporate the additional charged regions without lightning because the charge structures were determined by the temporal changes in lightning discharges[33]. In this study, the inversion of multi-point E-field data provides an effective tool for characterizing the finer charge structure of dust storms. The inversion results suggest that the mosaic charge structure is a general feature of dust storms, which can be explained by the different responses of oppositely charged particles to turbulence. For dust storms #1–#3, the dust particles collected at 5 m height showed that particle sizes varied widely from ~1 $\mu$m to ~300 $\mu$m (Supplementary Fig. 7b and c).

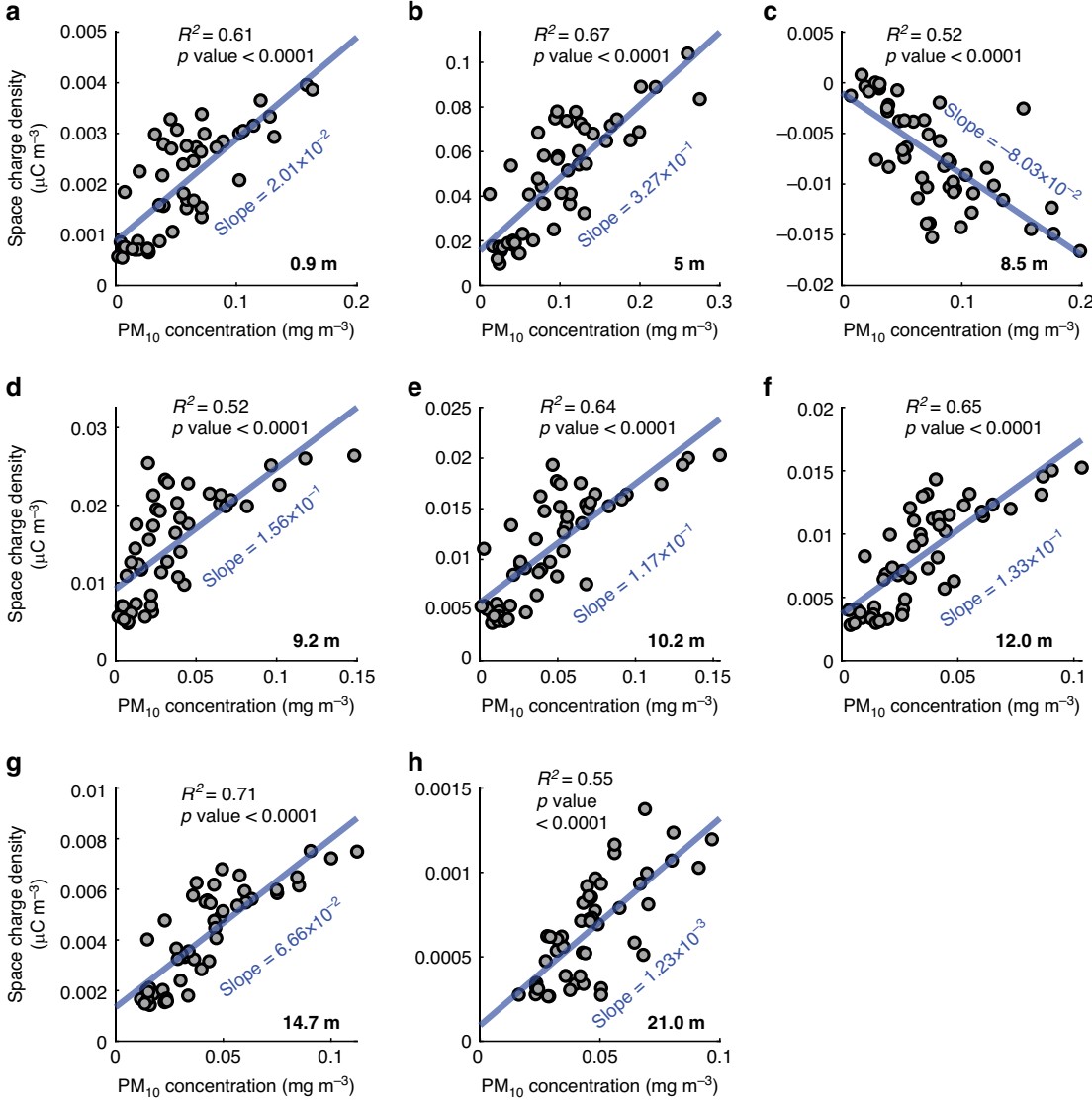

**Fig. 7 Significant linear relationships between the $PM_{10}$ concentrations and space-charge densities for storm #1. a–h** Linear relationships at the heights from 0.9 m to 21 m. The $PM_{10}$ concentration data for 30 m was not available during storm #1. Here, symbols denote the reconstructed space-charge density vs. the $2^9$ s time-varying mean of the measured $PM_{10}$ concentration (extracted by the discrete wavelet transform), and lines denote linear regressions (coefficient of determination $R^2$ and $p$ value are shown). For these data, the ambient temperature and relative humidity are in the range of $16.7 \pm 1.2\,°C$ and $17.4 \pm 3.3\%$, respectively.

On the basis of measured wind velocity data, the Stokes number $S_t$ of dust particles, which is defined as the ratio of the particle relaxation timescale and the Kolmogorov timescale[59], is estimated to lie in the range $\mathcal{O}(10^{-3}) - \mathcal{O}(10^2)$ (see Methods and Supplementary Fig. 20). In fact, numerical simulations[43,45] and laboratory experiments[44] have demonstrated that, in particle-laden turbulent flows, significant charge separations can be caused by turbulence. As previously demonstrated, the negatively charged smaller particles with $S_t \lesssim \mathcal{O}(1)$ could preferentially accumulate in the high-strain-rate regions of the wind flow due to turbulence, while the positively charged larger particles with $S_t \gtrsim \mathcal{O}(1)$ may be more uniformly distributed than smaller particles[43–45]. Since dust storms are typically polydisperse particle-laden turbulent flows at very-high-Reynolds-number, we can reasonably speculate that charge separations by turbulence are prevalent in dust storms, thereby leading to a general 3D mosaic charge structure. In this study, because the flow conditions were almost the same in the three dust storms, in general, it is unsurprising that these dust storms would exhibit a very similar vortex structure, so that the

charged particles driven by such similar vortex structures could form very similar charge structure.

In summary, on the basis of E-field data measured at the QLOA, an inversion method is proposed to estimate the space-charge density and E-field of dust storms. The collected E-field data were obtained from 19 components distributed over a region of $20 \times 20 \times 9\ m^3$ with a spacing of 10 m, 5 m, and 1.75 m in the $x$-, $y$-, and $z$-direction, respectively. The inversion method was based on a Fredholm integral equation of the first kind and combined with the standard Tikhonov regularization that requires the 2-norm of the solution residual to be small. The reconstructed results obtained from the selected high-quality data agree well with the measured data (with a mean residual of about 0.04) and the GLA-based density. The observed three dust storms exhibited a general mosaic charge structure, which is likely due to the separation of oppositely charged dust particles by turbulence. Next, the E-field can be predicted by the Coulomb's law based on the reconstructed space-charge density. In addition, we find that the large-scale electrification dynamic equilibrium consistently

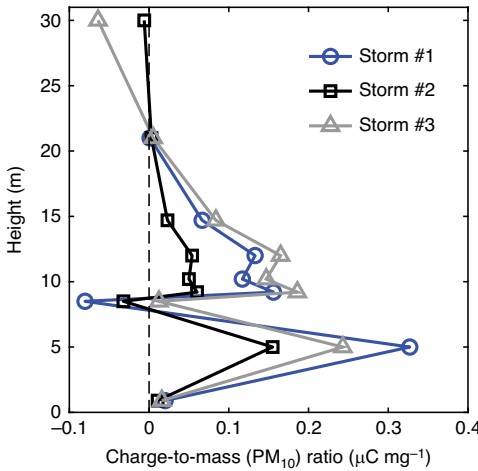

**Fig. 8 Vertical profiles of the charge-to-mass (PM₁₀) ratios for storms #1-#3.** For storm #1, the ambient temperature ($T$) and relative humidity (RH) are in the range of 16.7 ± 1.2 °C and 17.4 ± 3.3%, respectively. For storm #2, the ambient $T$ and RH are in the range of 20.2 ± 0.5 °C and 20.1 ± 0.9%, respectively. For storm #3, the ambient $T$ and RH are in the range of 9.0 ± 1.4 °C and 15.6 ± 2.5%, respectively.

exists at different heights, suggesting that the charge-to-mass (PM₁₀) ratios remain constant at specific height but vary with height in both magnitude and polarity. This study provides the quantitative insight into the 3D electrical structure of dust storms, which is underexplored due to the limitations of the currently used measurement techniques. Our method can be also an effective and reliable tool in exploring the electrical properties of other granular systems, such as fluidized beds, dust devils, blowing snows, volcanic eruptions, etc.

## Methods

**Discretization of the forward model.** As previously mentioned, the forward problem [i.e., Eqs. (1) and (2)] can be theoretically expressed as

$$\mathbf{E}(x, y, z) = \iiint_{\Omega} \mathbf{K}(\mathbf{r}, \widetilde{\mathbf{r}})\rho(\boldsymbol{v})\mathrm{d}^3\boldsymbol{v}, \tag{5}$$

where $\mathbf{E}(x, y, z)$ is the measured E-fields at point $(x, y, z)$, $\Omega$ is the computational domain, $\mathbf{r} = (x - x', y - y', z - z')$, $\widetilde{\mathbf{r}} = (x - x', y - y', z + z')$, and $\boldsymbol{v} = (x', y', z')$. In practice, since the number of measurement points is finite, the forward and inverse problems, Eq. (5), are usually solved in the space of discrete data and model parameters.

We use the Galerkin method to discretize Eq. (5) with $n$ orthonormal boxcar basis functions[48,53]:

$$\psi_i(\boldsymbol{v}) = \begin{cases} 1, & \boldsymbol{v} \in \Omega_i, \\ 0, & \boldsymbol{v} \notin \Omega_i, \end{cases} \tag{6}$$

where the domain $\Omega$ is divided into $n = 400 \times 400 \times 60$ small nonoverlapping rectangular cells $\Omega_i$ (i.e., $\Omega = \bigcup_{i=1}^{n} \Omega_i$). Thus, the space-charge density $\rho(\boldsymbol{v})$ can be approximated by its projection over the boxcar basis functions $\psi_i(\boldsymbol{v})$, that is

$$\rho(\boldsymbol{v}) \approx \sum_{i=1}^{n} a_i \psi_i(\boldsymbol{v}), \tag{7}$$

where $a_i$ are the unknown coefficients of the series expansion. Clearly, such a finite expansion of $\rho(\boldsymbol{v})$ by Eq. (7) is impossible to satisfy Eq. (5) exactly. Substituting Eq. (7) into Eq. (5), and according to the rules of sum and additivity of triple integrals, produces the following residual or error $\mathbf{E}^{\mathrm{R}}(x, y, z)$ of the integral equation, Eq. (5):

$$\mathbf{E}^{\mathrm{R}}(x, y, z) = \mathbf{E}(x, y, z) - \sum_{i=1}^{n} a_i \iiint_{\Omega_i} \mathbf{K}(\mathbf{r}, \widetilde{\mathbf{r}})\mathrm{d}^3\boldsymbol{v} \tag{8}$$

According to the Galerkin method, we use the original boxcar basis functions $\psi_j(\boldsymbol{v})$ as the weighting functions to make the following weighted integrals of

residuals $\mathbf{E}^{\mathrm{R}}(x, y, z)$ equal to zero:

$$\iiint_{\Omega} \mathbf{E}(x, y, z)\psi_j(\boldsymbol{v})\mathrm{d}^3\boldsymbol{v} - \sum_{i=1}^{n} a_i \iiint_{\Omega} \psi_j(\boldsymbol{v}) \left[ \iiint_{\Omega_i} \mathbf{K}(\mathbf{r}, \widetilde{\mathbf{r}})\mathrm{d}^3\boldsymbol{v} \right] \mathrm{d}^3\boldsymbol{v} = 0$$
$$(j = 1, 2, \cdots, m) \tag{9}$$

Here, $m = 19$ is the number of measurement components used in inversions, except for the case of subsampling inversion where $m$ is less than 19. According to Eq. (6), Eq. (9) can be readily simplified as

$$\mathbf{E}(x_j, y_j, z_j) = \sum_{i=1}^{n} a_i \iiint_{\Omega_i} \mathbf{K}(\mathbf{r}_j, \widetilde{\mathbf{r}}_j)d^3\boldsymbol{v}, \quad j = 1, 2, ..., m \tag{10}$$

where $(x_j, y_j, z_j)$ are the coordinates of the measurement points, $\mathbf{r}_j = (x_j - x', y_j - y', z_j - z')$, and $\widetilde{\mathbf{r}}_j = (x_j - x', y_j - y', z_j + z')$. We can write Eq. (10) in a matrix form:

$$\begin{bmatrix} G_{1,1} & G_{1,2} & \dots & G_{1,n} \\ & \ddots & \vdots & \\ G_{m,1} & G_{m,2} & \dots & G_{m,n} \end{bmatrix} \begin{bmatrix} a_1 \\ \vdots \\ a_n \end{bmatrix} = \begin{bmatrix} E(x_1, y_1, z_1) \\ \vdots \\ E(x_m, y_m, z_m) \end{bmatrix} \tag{11}$$

with the entry as

$$G_{j,i} = \iiint_{\Omega_i} \mathbf{K}(\mathbf{r}_j, \widetilde{\mathbf{r}}_j)\mathrm{d}^3\boldsymbol{v} \tag{12}$$

which can be numerically evaluated by Gaussian Quadrature. Since the linear systems of algebraic equations [i.e., Eq. (11)] are ill-posed, the coefficients $a_i$ should be determined by solving the constrained optimization problem [i.e., described by Eq. (4)], as discussed in detail in the following section.

**Solving the inverse space-charge problem.** The inverse problem [i.e., Eq. (4)] is solved following a method based on the singular value decomposition of matrix $\mathbf{G}$:

$$\mathbf{G} = \mathbf{U}\boldsymbol{\Sigma}\mathbf{V}^T, \tag{13}$$

where $\mathbf{U}$ and $\mathbf{V}$ are orthogonal unitary matrices whose columns are the left singular vectors $u_i$ and the right singular vectors $v_i$, respectively; and $\boldsymbol{\Sigma} = \mathrm{diag}(\sigma_1, \sigma_2, \cdots, \sigma_N)$ consists of the singular values of $\mathbf{G}$ sorted in descending order. Since the ill-posedness of the inverse problem is largely due to the small singular values $\sigma_i$, the idea of the regularization is to filter out the solution corresponding to the small $\sigma_i$. Therefore, the inverted (or regularized) solution of the space-charge density $\boldsymbol{\rho}_{\mathrm{inv}}$ can be expressed as follows[52,53]

$$\boldsymbol{\rho}_{\mathrm{inv}} = \mathbf{V}\mathbf{S}^{-}\mathbf{U}^T\mathbf{E}^{\mathrm{obs}} \tag{14}$$

where $S_{ij}^{-} = \sigma_i\delta_{ij}/(\sigma_i^2 + \lambda_{\mathrm{opt}}^2)$; $\delta_{ij}$ is the Kronecker delta, i.e., if $i = j$, $\delta_{ij} = 1$, and if $i \neq j$, $\delta_{ij} = 0$.

It is clear that the key question for solving the inverse problem is to make a good selection of the optimal regularization parameter $\lambda$, because it represents the trade-off between the misfit and regularization. In this study, we employed the generalized cross-validation (GCV) method to select an optimal value of $\lambda$[60,61]. The optimal value $\lambda_{\mathrm{opt}}$ is the minimum point of the GCV function:

$$\mathcal{G}(\lambda) = \frac{\left\| \mathbf{G}\boldsymbol{\rho}_{\mathrm{inv}} - \mathbf{E}^{\mathrm{obs}} \right\|_{\mathbb{L}_2}^2}{\left[ \mathrm{trace}(\mathbf{I}_m - \mathbf{G}\mathbf{G}^{\#}) \right]^2}, \tag{15}$$

where $\mathbf{I}_m \in \mathbb{R}^{m \times m}$ is an identity matrix, and $\mathbf{G}^{\#} \in \mathbb{R}^{n \times m}$ is a matrix that produces the regularized solution, i.e., $\boldsymbol{\rho}_{\mathrm{inv}} = \mathbf{G}^{\#}\mathbf{E}^{\mathrm{obs}}$.

The inversion domain $\Omega$ was extended for 2 km in the $x$-direction ($L_x = 1$ km), 1 km in the $y$-direction ($L_y = 0.5$ km), and 0.3 km ($L_z = 0.3$ km) in the $z$-direction. The vertical size of the inversion domain was determined based on the dust concentration measurements, where PM₁₀ concentration decreased exponentially with height[50,62] and reached zero at the height of well below 0.2 km (Supplementary Figs. 4–6). We constructed a numerical grid with a size of $5 \times 2.5$ m² in the $x$- and $y$-direction, and a grid stretching parameter of 1.1 in the $z$-direction, resulting in an increase in the vertical grid size from 0.1 to ~27.7 m. There was a total of 9,600,000 grid cells.

**Extraction of time-varying mean by the discrete wavelet transform.** We use the discrete wavelet transform method to extract the time-varying means of the measured E-field and PM₁₀ series over the $2^9$ s timescales. The discrete wavelet transform is performed by the Daubechies wavelet[63] of order 10 (i.e., db10) at level 9, and thus the data series $X$ with sampling interval $\Delta t$ can be decomposed into[64]:

$$X = \sum_{i=1}^{9} D_i + S_9 \tag{16}$$

Here, $D_i$ is referred to as the $i$th level wavelet detail, which represents the changes of $X$ on a scale of $2^{i-1}\Delta t$ s; and $S_9$ is referred to as the 9th level wavelet smooth or approximation of $X$, which represents the means of $X$ over a scale of $2^9\Delta t$ s. In this

study, the sampling intervals $\Delta t$ are one second for the E-field and $PM_{10}$ measurements. Thus, $S_9$ can be considered as an approximation to the time-varying mean of $X$ series over the $2^9$ s timescale, and the 9th level wavelet rough $\sum_{i=1}^{9} D_i$ can be regarded as the fluctuation of $X$ series[64].

**Residual analysis**. To assess the accuracy of inversion, the normalized residual $\zeta$ (also known as the squared relative $\mathbb{L}_2$ error) between the observed data and the model prediction was proposed and can be defined by[65,66]

$$\zeta = \frac{\left\| \mathbf{G} \boldsymbol{\rho}_{\text{inv}} - \mathbf{E}^{\text{obs}} \right\|_{\mathbb{L}_2}^2}{\left\| \mathbf{E}^{\text{obs}} \right\|_{\mathbb{L}_2}^2} \quad (17)$$

The smaller the value of $\zeta$, the better correspondence between the model and observed data. If $\zeta$ is zero, the model fits the data perfectly.

**GLA-based space-charge density**. According to Gauss's law, it is known that the space-charge density at one point is proportional to the divergence of the E-field of this point, which allows us to estimate $\rho$ indirectly by measuring E-field divergence. In such estimations, the spatial derivatives with respect to three orthogonal coordinates of the E-field at a measurement point are needed. As shown in Fig. 1c and Supplementary Table 1, E-field measurements along three orthogonal coordinates were conducted only at p9 point in our observation array. In this case, the GLA-based density $\rho_{\text{GLA}}$ can be only determined at point p9 by[4,27,30]

$$\rho_{\text{GLA}} = \varepsilon_0 \nabla \cdot \mathbf{E} \quad (18)$$

Here, we use the spline-interpolation method to evaluate the partial derivatives $\partial E_x / \partial x$, $\partial E_y / \partial y$, and $\partial E_z / \partial z$ based on the measured data (see ref. [27] for more details).

**Estimating particle's stokes number**. By definition, the Stokes number $S_t$ is defined as the ratio of the particle relaxation timescale $\tau_p$ and the Kolmogorov timescale $\tau_\eta$, such that

$$S_t = \frac{\tau_p}{\tau_\eta}. \quad (19)$$

In the typical cases that particle Reynolds number is <1 and particles are much denser than the fluid[50,67], particle relaxation timescale can be estimated by[59]

$$\tau_p = \frac{\rho_p d_p^2}{18 \nu \rho_a}, \quad (20)$$

where $\rho_p$ and $\rho_a$ are particle and fluid mass density, respectively; $d_p$ is particle diameter; $\nu$ is the kinematic viscosity of the fluid. In the log-law region, the Kolmogorov timescale $\tau_\eta$ can be estimated by the flowing equations[68]

$$\begin{cases} \tau_\eta = \frac{\eta^2}{\nu} \\ \frac{\eta}{\delta_\nu} = (\kappa z^+)^{1/4} \end{cases}, \quad (21)$$

where $\eta$ is the Kolmogorov microscale, $\kappa = 0.41$ is the Von Kármán constant, $\delta_\nu = \nu / u_\tau$ is the viscous lengthscale, $u_\tau$ is the friction velocity, $z^+ = z / \delta_\nu$ is the dimensionless height measured in viscous lengthscale. Based on the measured wind velocity at p9, the $S_t$ number of dust particles can be estimated using Eqs. (19)–(21).

## Data availability
The E-field data used to perform the inverse calculations of storms #1-#3 are available at https://doi.org/10.6084/m9.figshare.12866744.

## Code availability
The code used to perform inverse calculations is available upon reasonable request to the corresponding author.

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

## Acknowledgements

This work was supported by the National Natural Science Foundation of China (No. 11802109), the Young Elite Scientists Sponsorship Program by CAST (No. 2017QNRC001), and the Fundamental Research Funds for the Central Universities (No. lzujbky-2018-7).

## Author contributions

Y.H.Z. designed and organized the research and its approach, as well as analyzed the results. H.Z. carried out the field observations, analyzed the data, and performed the inverse calculations. All authors contributed to the paper.

## Competing interests

The authors declare no competing interests.
