## [Peer Review File · Nature Communications]

Reviewers' Comments:

Reviewer #1:

Remarks to the Author:

Review of "Reconstructing the electrical structure of dust storms from locally observed electric field data" by Zhang et al.

General comments:

The authors have described an inversion method to infer the charge structure of a dilute granular flow based on multiple ground-based electric field, particle, and meteorological measurements. The principal conclusion of the work is that a multitude of localized 3D charge structures exist in dust storms that vary with particle concentration. The methods of studying electrified granular systems remotely presented here likely has applications in many fields. However, the authors ignore to acknowledge important pieces of geophysical research that have previously reached similar conclusions regarding the charge structure of airborne granular flows. These studies primarily come from the volcanological context. While volcanic plumes and dust storms are undoubtedly underscored by different dynamics, they share many similarities like scale, particle compositions (silicates), and electrification mechanisms. Furthermore, many of the instruments used to characterize electrostatic processes in volcanic plumes are identical to those used in duststorms (e.g. field mills). Thus, while the inversion method presented in Zhang et al. is certainly novel and exciting, the scientific conclusions reached by authors are somewhat less novel.

1) 3D structure of large-scale granular flows

The authors mention that previous measurements of airborne granular flows have been essentially 1D in nature. This is not true. Behnke and others, for instance, have inverted the 3D charge structure in numerous large scale silicate particle clouds using ground based sensors (see, for instance, Behnke et al., 2014).

2) Charge mosaic

A multitude of concentrated regions of charge have been inferred from Lightning Mapping Arrays at number of volcanoes. Indeed, Aizawa note that the character of lightning in the volcanic columns at Sakurajima can only be explained by a mosaic of charge. Similar mosaics of charge were described by Woodhouse and Behnke in an ash cloud during the 2010 eruption of Eyjafjallajökull.

1) Separation of charge due to turbulence:

The authors note that charge clustering in dust storms likely results from the fact that positive charge is carried on large grains while smaller grain concentrate negative charge. Smaller particles (and negative charges) become concentrated in in turbulent eddies. Such inferences were made experimentally by Cimarelli et al 2014.

Again, while volcanic plumes and dust storms cannot be compared on a one to one basis, information gained from studying one system likely provides insight into mechanisms operating in the other. Zhang et al present their results as if they were the first of their kind. This is simply not true and does not recognize the efforts of other investigators over the last 20 years!

In conclusion, I think this work is worth publishing, but I do not think that the scientific claims the authors make are as novel as they suggest.

Other comments:

Triboelectric charging should be discussed in introduction, not results.

"Because the mosaic charge structure could be attributed to the effects of turbulence, as stated by Renzo and Urzay" is not a sentence.

Reviewer #2:

Remarks to the Author:

The paper is related to the mutual effect of dust dynamics on particle charging that is a topic of interest for a wide community with applications in several fields of investigation.

The focus here is in the behavior of airborne dust and in the electrification process that occur during dust events like dust storms. The authors reconstruct the space-charge density structure of dust storms starting from a mathematical inversion model applied on experimental data. This is absolutely a novel and highly interesting result that, if well proven, may improve our understanding on the physics of dust lifting, electrification and transportation.

Anyway, in the present stage, the paper needs some major improvements in order to prove the reliability of the presented results. If the paper outcome will be well proven, it will be surely suitable for publishing in Nature Communication.

Major revisions:

1) Considering that the VREFM is not a commercial instrument, the authors should give some more details about its working principle and the setup of the experiment. How is the instrument used to measure the 3D E-field? Probably the instrument is simply oriented in the x, y or z direction to acquire E_x , E_y , E_z . In that case, how do authors screen it from the windblown flux of charged particles? In the used setup, could the charged particles hit the sensing plate? This would alter the measurement results. Please, clarify.

2) It is really surprising to me that the authors find very similar mosaic structures in all the three presented dust storms. If the segregation of charged particles with similar sign is driven by the turbulence, how do authors explain the finding shown in Figs 4 and 4-5 of Supplementary material? These plots show that clusters of particles with the same charge sign distribute always in the same spatial regions, at the same height, in all phases of a single storm and more or less in the same way for all the presented storms. How to explain a similar regularity and reproducibility in turbulent flows?

3) Moreover, in order to be sure that the resulted mosaic structures are not an artifact due to the constrained mathematical method used to model the data and/or to the used grid of sampled data, I think the calculations should be repeated by using data obtained from a sub-sample of random selected measurement points. A comparison between the 3D structure of space-charge density obtained with this sub-sample vs the complete sample could help clarify this point.

4) Results show a 3D structure of the space-charge density during dust storms. Anyway, you got 3D E-field data only in the p9 measurement point. How does this impact the quality of the results?

5) Correlation between space charge density and PM10 concentration is very interesting. Some of the plots in Fig. 6 and Supplementary Figs. 8 and 9 show an inversion in the trend with negative slopes at specific heights (8.5 m and 30 m). How do authors explain this finding?

Minor revisions:

- Fig. 1 caption: change "squares" with "triangles".
- Fig. 2 should be enlarged for a clearer view.

- Raw 152, Fig. 3a,b,c: Authors stated that "The residuals show no deterministic trend". Anyway, Figs 3a, 3b and 3c seem to show a slightly increase of normalized residuals vs time. Please, comment.

Reviewer #3:

Remarks to the Author:

Review of: "Reconstructing the electrical structure of dust storms from locally observed electric field data" by Zhang and Zhou [NCOMMS-20-04698]

The paper presents model results aiming to elucidate the complex electrical structure of dust storms. This is an important contribution that presents a significant progress in the field. The paper is very lucid, clear and well organized. The graphs are mostly adequate but require improvement. However, there are a few issues that require the authors' attention before the manuscript can be considered acceptable for publication.

Major Comments

1. The authors present 3 dust storm case studies that differ in their electrical behavior. However, no mention is made about the meteorological circumstances of these 3 events, particularly the back-trajectory of the winds that can indicate the source of the dust. Is it coming from the same source? Can the authors add information on the mineralogy of the particles? It is known that different substances have various dielectric constants, and so the fixed value $k=5$ used in their calculations may actually mask differences between the storms.
2. When computing the space charge density and the mosaic structure, what is obviously missing is the average volume charge (pC / m^3) and charge-per-mass (as in their 2013 GRL paper), quantities that testifies to the level of electrification of single particles. This can be easily achieved by dividing the total charge by the aerosol number concentration. Then, compare your results (presented in Figure 5) to the values obtained by Nicoll et al. (2011) for the Saharan Dust Layer (SAL) where they showed that dust plumes carried westward from the Sahara Desert are electrified with maximum charge densities of $\sim 5\text{-}25 \text{ pC m}^{-3}$.
3. The verification of the model results against the observations are presented only for p9. This seems arbitrary, and should be explained and justified. The point itself is not marked on Figure 1b (at least I could not find it, perhaps the figure is too small). The coordinates of p9 in terms of height above the surface are found in Table 1 ($z=5 \text{ m}$), but should be clarified in the text as well because the results refer to this point specifically.
4. The sensitivity of model performance to the selected point should be presented and discussed. How did the model perform for p1 or p2? Does distance or height have any effect on the residuals obtained from the GLA-based space charge density? This reviewer believes that at least one such comparison should be conducted and its results analyzed and discussed.

Graphics Comments

In general, the graphics are very good and clear, but some improvements are needed.

1. Figure 1: Please enlarge, if possible. There are so many details that one has to make a real effort in order to look for specific features. he
2. Figure 3: (a) For the middle graphs, I think that it would be better to list the heights of the sensors and not their names on the x-axis ($s_1=5\text{m}$; $s_5=5 \text{ m}$ etc.). The way it is presented now is quite meaningless (b) The time-line of the x-axis in the bottom graphs is inconvenient (huge numbers of seconds). Instead of using cumulative seconds, its much better to convert to minutes or better yet, to actual time, such that a comparison can be made to Figure 2. Why choose different times from the different storms? (give rational for this).
3. Figures 4 and 5 (and Supplementary Figure 7) are valuable and exhibit the results in a clear manner. Two suggestions here: (a) Enlarge by $\sim 15\%$ (if space allows) and (b) rotate the viewing

angle such that the vertical cross-section on the z-axis is much easier to discern. This can be easily achieved, and will support the main new finding of the mosaic structure.

Minor Comments

1. Line 119: For the assumption of a planar dielectric sandy ground to apply, the authors need to ascertain that the entire terrain in the immediate surroundings of their instrument is indeed uniform and does not exhibit any change in physical properties. Is this indeed so at the location?
2. Paragraph from line 93-104: how was the storms' intensity determined? Was this just a function of the wind speed, the PM10 concentration or the enhancement of the electric field? In line 101, what does "rapidly" and "immediately" mean? Please give a quantitative measure (minutes or seconds).
3. Line 192: Equilibrium in charge distribution on a poly-dispersed aerosol distribution had been described by Hoppel and Frick (JGR, 1986) and Yair and Levin (JGR, 1989). These references should be included.
4. Line 196: What do you mean by synchronous evolution? This term implies coupling, not just mere temporal coincidence.
5. Line 230: what does "high-stain-rate regions" mean? Please explain.
6. Line 246: consider deleting the word "easily". I believe that it is a substantial effort.
7. Lines 218, 220: Recommend adding references to recent works on dust storms, and addressing their findings in the context of the discussion:

Silva, H.G., Lopes, F.M., Pererira, S., Nicoll, K., Barbosa, S.M., Conceicao, R., Neves, S., Harrison, R.G., Pereira, M.C. (2016). Saharan dust electrification perceived by a triangle of atmospheric electricity stations in Southern Portugal. *J. Electro.* 84, 106–120.

Katz, S. Y. Yair, C. Price, R. Yaniv, I. Silber, B. Lynn and B. Ziv (2018), Electrical properties of the 8-12th September, 2015 massive dust outbreak over the Levant. *Atmos. Res.*, 201, 218-225.

Solomos, S., Ansmann, A., Mamouri, R.-E., Biniotogoulou, I., Patlakas, P., Marinou, E., Amiridis, V., (2017). Remote sensing and modeling analysis of the extreme dust storm hitting the Middle east and eastern Mediterranean in September 2015. *Atmos. Chem. Phys.* 17, 4063–4079.

Yair, Y., S. Katz, R. Yaniv, B. Ziv and C. Price (2016), An electrified dust storm over the Negev desert, Israel. *Atmos. Res.*, 181, 6-71

We thank the reviewers for the constructive comments and suggestions. We have substantially modified the manuscript and included more supplementary information accordingly. We believe that these modifications have considerably improved the manuscript. The following text contains the reviewers' comments (in black), our responses (in blue). All modifications in the revised manuscript associated with the reviewers' comments are highlighted in blue.

Reviewer #1 (Remarks to the Author):

Review of "Reconstructing the electrical structure of dust storms from locally observed electric field data" by Zhang et al.

General comments: The authors have described an inversion method to infer the charge structure of a dilute granular flow based on multiple ground-based electric field, particle, and meteorological measurements. The principal conclusion of the work is that a multitude of localized 3D charge structures exist in dust storms that vary with particle concentration. The methods of studying electrified granular systems remotely presented here likely has applications in many fields. However, the authors ignore to acknowledge important pieces of geophysical research that have previously reached similar conclusions regarding the charge structure of airborne granular flows. These studies primarily come from the volcanological context. While volcanic plumes and dust storms are undoubtedly underscored by different dynamics, they share many similarities like scale, particle compositions (silicates), and electrification mechanisms. Furthermore, many of the instruments used to characterize electrostatic processes in volcanic plumes are identical to those used in dust storms (e.g. field mills). Thus, while the inversion method presented in Zhang et al. is certainly novel and exciting, the scientific conclusions reached by authors are somewhat less novel.

Response: We thank the reviewer for this positive comment on our inversion method and providing important information about volcanic eruptions. We sincerely apologize for missing out these important studies that are highly relevant to our work. In the original manuscript, since we focused on the electrification of dust storms, we did not discuss the literatures in the field of volcanological context. We agree with the reviewer that dust storms and volcanic eruptions indeed share similar charge segregation mechanisms. For example, based on electric field measurement data, Williams et al. (2009) inferred that the charge structure of dust storms was monopolar or bipolar, which is also inferred in volcanic plumes based on the 3D lightning data (Behnke et al., 2014; Woodhouse & Behnke, 2014). In the revised manuscript, we have added a more detailed discussion of the electrification of volcanic eruptions. Please see lines 30, 39-41, 70-71, 304-306, 308-310, and 319-324 in the revised manuscript.

1) 3D structure of large-scale granular flows

The authors mention that previous measurements of airborne granular flows have been essentially 1D in nature. This is not true. Behnke and others, for instance, have inverted the 3D charge structure in numerous large scale silicate particle clouds using ground based sensors (see, for instance, Behnke et al., 2014).

Response: We thank the reviewer for bringing these important studies to our attention. Indeed, Woodhouse & Behnke (2014) and Behnke et al. (2014) have inferred the charge structures of volcanic plumes based on three-dimensional lightning data. In the revised manuscript, we have added this information in the Introduction and Discussion sections. Please see lines 30, 70-71, 255-257, 304-306, 308-310, and Refs. 32 and 33 in the revised manuscript.

2) Charge mosaic

A multitude of concentrated regions of charge have been inferred from Lightning Mapping Arrays at number of volcanoes. Indeed, Aizawa note that the character of lightning in the volcanic columns at Sakurajima can only be explained by a mosaic of charge. Similar mosaics of charge were described by Woodhouse and Behnke in an ash cloud during the 2010 eruption of Eyjafjallajökull.

Response: We thank the reviewer for bringing these relevant references to our attention. We agree with the reviewer that the mosaic charge pattern is a general feature of the disperse two-phase turbulent flows, such as volcanic plumes and dust storms. As suggested, we have added and discussed these studies (i.e. Aizawa et al., 2010; Woodhouse & Behnke, 2014) in the Results and Discussion sections in the revised manuscript. Please see lines 30, 70-71, 255-257, 304-306, 308-310, and Refs. 31 and 33 in the revised manuscript.

3) Separation of charge due to turbulence:

The authors note that charge clustering in dust storms likely results from the fact that positive charge is carried on large grains while smaller grain concentrate negative charge. Smaller particles (and negative charges) become concentrated in in turbulent eddies. Such inferences were made experimentally by Cimorelli et al 2014.

Response: We thank the reviewer for providing this very important reference. In this study, we infer that the mosaic charge pattern of dust storms is likely due to the charge segregation by turbulence. Importantly, Cimorelli et al. (2014) have provided direct experimental evidence for such inference. As suggested, we have added a more detailed discussion about Cimorelli et al. (2014) in the revised manuscript. Please see lines 75-77, 255-257, 319-324, and Ref. 44 in the revised manuscript.

Again, while volcanic plumes and dust storms cannot be compared on a one to one basis, information gained from studying one system likely provides insight into mechanisms operating in the other. Zhang et al present their results as if they were

the first of their kind. This is simply not true and does not recognize the efforts of other investigators over the last 20 years!

Response: We thank the reviewer again for pointing out this issue. Indeed, volcanic plumes and dust storms cannot be completely viewed as two independent systems, because under certain conditions the charge segregation mechanisms between these two flows are similar. We sincerely apologize for ignoring the very important studies about volcanic eruptions. As mentioned above, we have now added a more detailed discussion about electrical phenomena in volcanic eruptions, including charge structures and charge segregation mechanisms.

In conclusion, I think this work is worth publishing, but I do not think that the scientific claims the authors make are as novel as they suggest.

Response: We sincerely thank the reviewer for the positive comment on our work, even though the studies about volcanic eruptions were ignored in the original manuscript. In the revised manuscript, we have removed the sentences which state that we reveal the mosaic charge structures for the first time.

Other comments:

Triboelectric charging should be discussed in introduction, not results.

Response: We thank the reviewer for this constructive suggestion. We are not sure whether we follow your question. We consider that “triboelectric charging” may refer to “large-scale charge equilibrium phenomenon”. As suggested, we have discussed the triboelectric charging (“large-scale charge equilibrium phenomenon”) in detail in the Introduction section of the revised manuscript. Please see lines 80-86 in the revised manuscript.

“Because the mosaic charge structure could be attributed to the effects of turbulence, as stated by Renzo and Urzay” is not a sentence.

Response: We thank the reviewer for pointing out this error. We have corrected and rewritten this sentence and carefully checked for further grammatical problems.

Reviewer #2 (Remarks to the Author):

The paper is related to the mutual effect of dust dynamics on particle charging that is a topic of interest for a wide community with applications in several fields of investigation. The focus here is in the behavior of airborne dust and in the electrification process that occur during dust events like dust storms. The authors reconstruct the space-charge density structure of dust storms starting from a

mathematical inversion model applied on experimental data. This is absolutely a novel and highly interesting result that, if well proven, may improve our understanding on the physics of dust lifting, electrification and transportation. Anyway, in the present stage, the paper needs some major improvements in order to prove the reliability of the presented results. If the paper outcome will be well proven, it will be surely suitable for publishing in Nature Communication.

Response: We do appreciate the reviewer for the positive and constructive comments. As suggested, we have added some new content and analyses in order to verify the reliability of our results. Please see the following responses for more details.

Major revisions:

1) Considering that the VREFM is not a commercial instrument, the authors should give some more details about its working principle and the setup of the experiment. How is the instrument used to measure the 3D E-field? Probably the instrument is simply oriented in the x, y or z direction to acquire E_x , E_y , E_z . In that case, how do authors screen it from the windblown flux of charged particles? In the used setup, could the charged particles hit the sensing plate? This would alter the measurement results. Please, clarify.

Response: We thank the reviewer for this important comment. We apologize for the lack of a clear description of VREFM in the previous version of the manuscript. In the revised manuscript, we have added a brief description of the working principle of the VREFM in Supplementary note 1. The working principle of the VREFM is based on the vibrating capacitor technique. As the electrode vibrates, it will charge and discharge periodically, leading to induced electric current in the VREFM. This current is then converted to an output voltage whose value is directly proportional to the ambient electric field component normal to the electrode surface. Thus, the VREFM can be used to measure the electric field component normal to the electrode surface. The exact proportionality constants between the output voltages and normal electric field components are determined by the calibration experiment using a large parallel-plate electric-field calibrator. As the reviewer pointed out, we measured the 3D electric field by simply mounting the VREFMs directed along the x-, y- and z-axis. In this case, the VREFM is inevitably subjected to the impacting of charged particles, but such disturbance caused by charged particles can be considerably eliminated. This is because the induced electric current is a periodic signal with a fixed high vibration frequency of 200 Hz, whereas the impacting electric current due to the charged particles is a random signal with a relatively low-frequency band and thus can be filtered out by the signal processing modules in VREFM. Please see lines 121-123 and Supplementary note 1 (pages 33-35) in the revised manuscript.

2) It is really surprising to me that the authors find very similar mosaic structures in all the three presented dust storms. If the segregation of charged particles with similar sign is driven by the turbulence, how do authors explain the finding shown in Figs 4

and 4-5 of Supplementary material? These plots show that clusters of particles with the same charge sign distribute always in the same spatial regions, at the same height, in all phases of a single storm and more or less in the same way for all the presented storms. How to explain a similar regularity and reproducibility in turbulent flows?

Response: We thank the reviewer for this important comment. We are sorry that this issue was not clear enough in the original version of the manuscript. In the revised manuscript, we have adequately explained why all the three dust storms exhibited a very similar mosaic structure at different stages. The main reasons are as follows:

As we mentioned in the original manuscript, the inversion results were performed using the time-varying mean data over the 2^9 s timescales, rather than raw data. This suggests that the reconstructed space-charge density is an average density over the 2^9 s timescales. In this case, the instantaneous changes in space-charge structure cannot be revealed, thus showing a very similar charge structure at different stages of each storm (e.g. Fig. 5 in the revised manuscript). If we use raw data to perform inversion, the instantaneous changes in space-charge structure are apparent (Supplementary Figs. 5-7). However, since the inversion performance is very sensitive to electric field fluctuations, such inversion with raw data could result in a failure of the inversion (e.g. a very large residual up to 0.35 in Supplementary Fig. 5b). This is because the small-scale (high-frequency) fluctuations of the electric field data at a measurement point are dominated by turbulence and are probably due to the local changes in space-charge densities. Such small-scale and local changes at a measurement point cannot be reflected at other points far from it. For these reasons, this study was mainly focused on the inversion with time-varying mean data and thus the results are relevant to the large-scale averaged electrical properties.

On the other hand, due to almost the same flow conditions and particle properties for the three observed dust storms, the charge patterns are undoubtedly expected to be very similar. In the revised manuscript, we have added the measurements of wind directions and the mineralogical compositions of the dust samples in Fig. 2 and Supplementary Fig. 4, respectively. As shown in Figs. 2c, 2f, and 2i, the wind directions of the three dust storms lay within $152.3 \pm 4.7^\circ$, $160.9 \pm 6.4^\circ$, and $171.4 \pm 7.2^\circ$, respectively, showing that all storms mostly originated from the Badain Juran Desert (Fig. 1a). Meanwhile, the same dust properties for the three dust storms was also verified by the very similar size distributions and mineralogical compositions of the dust samples collected at measurement point p9 (please see Supplementary Fig. 4).

In summary, the similar reconstructed mosaic structures in all the three observed dust storms at different stages can be explained by the following two factors: (1) the inversions were performed with the time-varying mean data, thus exhibiting the large-scale averaged electrical properties; (2) the wind conditions and the particle properties were almost the same among three dust storms. Please see lines 200-209 and 246-253 in the revised manuscript.

3) Moreover, in order to be sure that the resulted mosaic structures are not an artifact due to the constrained mathematical method used to model the data and/or to the used grid of sampled data, I think the calculations should be repeated by using data obtained from a sub-sample of random selected measurement points. A comparison between the 3D structure of space-charge density obtained with this sub-sample vs the complete sample could help clarify this point.

Response: We do thank the reviewer for this excellent suggestion. Following this suggestion, we have added a set of random subsampling inversions in the revised manuscript in order to ensure that our inversion results are reliable. The subsample data set E_m^{obs} (subsampling size $m < 19$) is randomly selected from the total 19 measurement points. As in Iglesias (2016) and Kriukova et al. (2017), we execute each subsampling inversion 10 times. Then, the reconstructed space-charge density and the relative error with respect to the original 19-point inversion were computed and averaged over the 10 trials at each subsampling inversion. As shown in Fig. 3 and Supplementary Figs. 8-10, the relative errors

$$\frac{\|\rho_{inv}^m - \rho_{inv}^{19}\|_{L_2}}{\|\rho_{inv}^{19}\|_{L_2}}$$

decrease rapidly with increasing m and reduced to ~ 0.1 for the three dust storms. In addition, there are almost no differences in charge patterns when m exceeds 17 for each dust storm. This suggests that the densities ρ_{inv} reconstructed from the complete 19-point measurement data are reasonable and reliable, where all the relative errors are less than 10 %. Please see lines 211-223 and Refs. 55 and 56 in the revised manuscript.

References:

- Iglesias, M. A. (2016). A regularizing iterative ensemble Kalman method for PDE-constrained inverse problems. *Inverse Problems*, 32(2), 025002.
- Kriukova, G., Pereverzyev Jr, S., & Tkachenko, P. (2017). Nyström type subsampling analyzed as a regularized projection. *Inverse Problems*, 33(7), 074001.

4) Results show a 3D structure of the space-charge density during dust storms. Anyway, you got 3D E-field data only in the p9 measurement point. How does this impact the quality of the results?

Response: We thank the reviewer for this important comment. The measurements with 3D E-field only at point p9 does not affect the inversion results considerably. The reasons are as follows. From Eqs. (1) and (2), we can see that each electric field component (E_x , E_y and E_z) is dependent on the distribution of the space charge density. This means that, to some extent, each component of electric field can reflect the whole charge distribution in the dust storms. Therefore, once we have sufficient number of measured electric field components distributed in a proper spatial extent, the charge structure can be well reconstructed from the measured data. In addition,

based on the results of subsampling inversion, we can reasonably conclude that such measurements with 3D data only at p9 have negligible effects on the inversion results, because the inversions with the 18-point data were expected to have relative L_2 error of approximately 0.1 with respect to 19-point inversion. Please see lines 211-223 in the revised manuscript.

In this study, the 3D electric field data at p9 is mainly used to calculate the GLA-based densities, because the partial derivatives $\partial E_x/\partial x$, $\partial E_y/\partial y$, and $\partial E_z/\partial z$ are evaluated based on the spline-interpolation method where the 3D electric field at p9 is necessary. Please see lines 426-430 in the revised manuscript.

5) Correlation between space charge density and PM10 concentration is very interesting. Some of the plots in Fig. 6 and Supplementary Figs. 8 and 9 show an inversion in the trend with negative slopes at specific heights (8.5 m and 30 m). How do authors explain this finding?

Response: We thank the reviewer for this important comment. In fact, the slope of the charge density vs. PM10 curve can be defined as a parameter termed charge-to-mass (PM10) ratio (having a unit of $\mu\text{C}/\text{mg}$), whose sign and magnitude represent the charge polarity and the level of electrification at the measurement point, respectively. The positive trend (i.e. positive slope) between the space charge density and PM10 indicates a positive charge density, while the negative trend (i.e. negative slope) at 8.5 m and 30 m indicates a negative charge density at these points. This suggests that the regions at different heights (for example, 5 m vs. 30 m) may be oppositely charged, consistent with our reconstructed mosaic charge structures where the charge polarity varies with height.

Minor revisions:

- Fig. 1 caption: change “squares” with “triangles”.

Response: We thank the reviewer for noticing this. We have now changed Fig. 1 and its caption and ensured that they are consistent.

- Fig. 2 should be enlarged for a clearer view.

Response: Fig. 2 has been enlarged for clarity in the revised manuscript.

- Raw 152, Fig. 3a,b,c: Authors stated that “The residuals show no deterministic trend”. Anyway, Figs 3a, 3b and 3c seem to show a slightly increase of normalized residuals vs time. Please, comment.

Response: As suggested, we have added a comment “For each storm, the residuals increase slightly with time, indicating that the long-period ambient noise and the instrument drift are negligible during measurements.” in the revised manuscript. Please see lines 229-231 in the revised manuscript.

Reviewer #3 (Remarks to the Author):

Review of: "Reconstructing the electrical structure of dust storms from locally observed electric field data" by Zhang and Zhou [NCOMMS-20-04698]

The paper presents model results aiming to elucidate the complex electrical structure of dust storms. This is an important contribution that presents a significant progress in the field. The paper is very lucid, clear and well organized. The graphs are mostly adequate but require improvement. However, there are a few issues that require the authors' attention before the manuscript can be considered acceptable for publication.

Response: We deeply appreciate the reviewer for the positive and constructive comments on our work. Below is a detailed response to the reviewer's comments.

Major Comments

1. The authors present 3 dust storm case studies that differ in their electrical behavior. However, no mention is made about the meteorological circumstances of these 3 events, particularly the back-trajectory of the winds that can indicate the source of the dust. Is it coming from the same source? Can the authors add information on the mineralogy of the particles? It is known that different substances have various dielectric constants, and so the fixed value $k=5$ used in their calculations may actually mask differences between the storms.

Response: We thank the reviewer for this very important suggestion. As suggested, we have added additional analyses of the wind directions and mineralogy of the collected dust particles. Yes, the three observed dust storms indeed originated from the same source area. As shown in Figs. 2c, 2f, and 2i, the wind directions of the three storms lay within $152.3 \pm 4.7^\circ$, $160.9 \pm 6.4^\circ$, and $171.4 \pm 7.2^\circ$, respectively. Hence, the all source areas for the three dust storms can be traced back to the Badain Juran Desert (see Fig. 1a). Meanwhile, we found that the collected dust samples at measurement points p9 had a very similar size distribution and mineralogical composition (see Supplementary Fig. 4). We thus reasonably assumed that the relative dielectric constants of the sandy ground, κ , in the three dust storms were almost identical for the same dust source and dry sandy ground. Please see lines 111-113 and 139-144 in the revised manuscript.

2. When computing the space charge density and the mosaic structure, what is obviously missing is the average volume charge (pc / m^3) and charge-per-mass (as in their 2013 GRL paper), quantities that testifies to the level of electrification of single particles. This can be easily achieved by dividing the total charge by the aerosol number concentration. Then, compare your results (presented in Figure 5) to the

values obtained by Nicoll et al. (2011) for the Saharan Dust Layer (SAL) where they showed that dust plumes carried westward from the Sahara Desert are electrified with maximum charge densities of $\sim 5\text{-}25 \text{ pC m}^{-3}$.

Response: We thank the reviewer for this important suggestion. In fact, the space-charge density and volume charge are exactly the same, because both of them represent the electric charge per unit volume. In this study, the ratio of reconstructed space-charge densities ρ_{inv} to PM10 concentration [termed charge-to-mass (PM10) ratio] at each measurement point can be used to measure the level of electrification of the observed three dust storms. The charge-to-mass (PM10) ratio is larger than the actual charge-to-mass ratio because we only measured the PM10 concentration rather than the mass of the total suspended particles. In the revised manuscript, we have added a more detailed discussion of charge-to-mass (PM10) ratio in subsection titled “Multi-point large-scale charge equilibrium” and Fig. 8. Please see lines 278-287 in the revised manuscript.

In addition, as shown in Figs. 4g-4i, the maximum reconstructed charge densities of the observed three dust storms is on the order of $\sim 0.4 \text{ }\mu\text{C m}^{-3}$, which is consistent with the measurements values of $\sim 0.01\text{-}0.1 \text{ }\mu\text{C m}^{-3}$ in dust storms by Kamra (Kamra1972) and dust devils by Crozier (Crozier, 1964) at approximately 1 m height, but is larger than the measurements values of $\sim 5\text{-}25 \text{ pC m}^{-3}$ in Saharan dust layer by Nicoll et al. (2011) at altitude up to 4 km. This is because the observed particle number concentration is low ($\sim 10 \text{ cm}^{-3}$) in Nicoll et al. (2011). However, in this study, the particle number concentration is very high because the visibility decreased to 0.09 km during storm #2. In the revised manuscript, we have now cited the reference, Nicoll et al. (2011), as suggested. Please see lines 235-240 and Refs. 15 and 20 in the revised manuscript.

3. The verification of the model results against the observations are presented only for p9. This seems arbitrary, and should be explained and justified. The point itself is not marked on Figure 1b (at least I could not find it, perhaps the figure is too small). The coordinates of p9 in terms of height above the surface are found in Table 1 ($z=5 \text{ m}$), but should be clarified in the text as well because the results refer to this point specifically.

Response: We thank the reviewer for this very important suggestion. In the revised manuscript, we verified our inversion method in three-fold: subsampling inversion and residual analysis with the entire 19-measurement electric field components, as well as comparison of the reconstructed densities with the GLA-based densities at point p9. In this study, we compare these two densities at p9 because the densities ρ_{GLA} can be only evaluated at point p9. The densities ρ_{GLA} are estimated based on the Gauss's law, that is, $\rho_{GLA} = \epsilon_0 \nabla \cdot E = \epsilon_0 (\partial E_x / \partial x + \partial E_y / \partial y + \partial E_z / \partial z)$. In this case, the spatial derivatives with respect to three orthogonal coordinates of the electric field (i.e. the gradient of the electric field) at a measurement point, $\partial E_x / \partial x$, $\partial E_y / \partial y$, and $\partial E_z / \partial z$, are needed to calculate ρ_{GLA} . As shown in Figs. 1b, 1c and Supplementary

Table 1, electric field measurements along three orthogonal coordinates were conducted only at p9 in our observation array. Please see lines 211-223, 426-430, and Fig. 1 in the revised manuscript.

Following the reviewer's suggestions, Fig. 1b has been enlarged for clarity and a new figure (i.e. Fig. 1c) has been added in order to clearly show the coordinates of each measurement point.

4. The sensitivity of model performance to the selected point should be presented and discussed. How did the model perform for p1 or p2? Does distance or height have any effect on the residuals obtained from the GLA-based space charge density? This reviewer believes that at least one such comparison should be conducted and its results analyzed and discussed.

Response: We thank the reviewer for this very important suggestion. In the revised manuscript, we have added this part in subsection titled "Verification of the inversion method". The sensitivity of model performance to the selected point are conducted by the random subsampling inversions. As shown in Fig. 3 and Supplementary Figs. 8-10, the relative errors

$$\frac{\|\rho_{inv}^m - \rho_{inv}^{19}\|_{L_2}}{\|\rho_{inv}^{19}\|_{L_2}}$$

decrease rapidly with increasing m and reduced to ~ 0.1 for the three dust storms. In addition, there are almost no differences in charge patterns when m exceeds 17 for each dust storm. If p1 (corresponds to s3 and s14) and p2 (corresponds to s2 and s13) points have been removed from the inversion data set E_m^{obs} (correspond to $m=15$), the relative L_2 error of the subsampling inversion could reach 0.4, suggesting that p1 and p2 points cannot be simultaneously removed from the inversion data set. Please see lines 211-223 and Refs. 55 and 56 in the revised manuscript.

Yes, distance (i.e. VREFM spacing) affects the upper limit of the estimation error of the GLA-based densities ρ_{GLA} at p9. As mentioned above, GLA-based densities ρ_{GLA} are estimated based on the Gauss's law, where the partial derivatives $\partial E_x/\partial x$, $\partial E_y/\partial y$, and $\partial E_z/\partial z$ are evaluated by the spline-interpolation method. First, the electric field distribution function is approximated by an interpolation function (i.e. natural cubic spline). Then, the partial derivative at p9 (for example, $\partial E_x/\partial x$) can be regard as the derivative of the natural cubic spline at p9. For such natural cubic spline approximating differentiation, the error is $\|s' - y'\| \leq h/\pi \|y''\|$, where s is the interpolating spline; y is the true data; $h = \max\{h_i\}$ is the maximum VREFM spacing (Hanke & Scherzer, 2001). We thus expect that a small VREFM spacing could lead to a lower upper limit of the estimation error of the GLA-based densities ρ_{GLA} . Because such error analysis has been already presented in our previous study, Zhang & Zheng (2018), we do not discuss it in this study any more.

References:

Hanke, M., & Scherzer, O. (2001). Inverse problems light: numerical

differentiation. *The American Mathematical Monthly*, 108(6), 512-521.

Graphics Comments

In general, the graphics are very good and clear, but some improvements are needed.

1. Figure 1: Please enlarge, if possible. There are so many details that one has to make a real effort in order to look for specific features.

Response: We thank the reviewer for this good suggestion. As suggested, Fig. 1 has been enlarged for clarity.

2. Figure 3: (a) For the middle graphs, I think that it would be better to list the heights of the sensors and not their names on the x-axis (s1=5m; s5=5 m etc.). The way it is presented now is quite meaningless (b) The time-line of the x-axis in the bottom graphs is inconvenient (huge numbers of seconds). Instead of using cumulative seconds, its much better to convert to minutes or better yet, to actual time, such that a comparison can be made to Figure 2. Why choose different times from the different storms? (give rational for this).

Response: We thank the reviewer for this good suggestion. As suggested, the sensor names in the middle graphs have been changed as their coordinates, and the x-axes of the bottom graphs have been changed as the "local time". We choose the time points where the storm intensities are nearly the largest to show the comparisons of the measured electric field data and the model predicted data. Please see Fig. 4 in the revised manuscript for more details.

3. Figures 4 and 5 (and Supplementary Figure 7) are valuable and exhibit the results in a clear manner. Two suggestions here: (a) Enlarge by ~15% (if space allows) and (b) rotate the viewing angle such that the vertical cross-section on the z-axis is much easier to discern. This can be easily achieved, and will support the main new finding of the mosaic structure.

Response: We thank the reviewer for this very important suggestion. As suggested, all figures associated with the structures of the space-charge density and electric field (for example, Figs. 5-6 in the revised manuscript) have been enlarged and rotated for clarity.

Minor Comments

1. Line 119: For the assumption of a planar dielectric sandy ground to apply, the authors need to ascertain that the entire terrain in the immediate surroundings of their instrument is indeed uniform and does not exhibit any change in physical properties. Is this indeed so at the location?

Response: We thank the reviewer for this important suggestion. Yes, the surrounding

sandy ground of the Qingtu Lake Observation Array is indeed uniform and does not exhibit any change in physical properties. Since the Qingtu Lake has been dry for about 60 years, the Qingtu Lake is currently a large dry lake whose flat-lakebed covers nearly 20 km². In addition, because the main dust source area of Qingtu Lake is the Badain Juran Desert, the physical properties of the sandy ground are reasonably expected to be almost uniform. Following the reviewer's suggestion, we have marked the flat-lakebed of Qingtu Lake in the inset of Fig. 1a in the revised manuscript.

2. Paragraph from line 93-104: how was the storms' intensity determined? Was this just a function of the wind speed, the PM10 concentration or the enhancement of the electric field? In line 101, what does "rapidly" and "immediately" mean? Please give a quantitative measure (minutes or seconds).

Response: We thank the reviewer for this comment. In general, the intensity of the dust storm is quantified by the visibility (Shao, 2008). When the visibility is less than 200 m, the dust storm can be referred to as a severe dust storm. It is generally accepted that visibility is highly related to the PM10 concentration and electric field intensity. That is, the larger the PM10 concentration, the lower the visibility and the higher the electric field. Thus, the storm intensity can be measured by the PM10 concentration and electric field intensity. Please see lines 147-148 in the revised manuscript.

In addition, "rapidly" means that the storm intensity increases to its maximum value within a relatively short time, and "immediately" means that after reaching its maximum the storm intensity begins to decrease without an obvious mature stage. Following the reviewer's suggestion, the related sentence has been changed as "During storm #3, the storm intensity increased to its maximum value within 1.5 hours then decreased without an obvious mature stage". Please see lines 153-155 in the revised manuscript.

3. Line 192: Equilibrium in charge distribution on a poly-dispersed aerosol distribution had been described by Hoppel and Frick (JGR, 1986) and Yair and Levin (JGR, 1989). These references should be included.

Response: We thank the reviewer for pointing out these very important studies. As suggested, we have now cited Hoppel & Frick (1986) and Yair & Levin (1989) in the revised manuscript. Please see Refs. 57-58.

4. Line 196: What do you mean by synchronous evolution? This term implies coupling, not just mere temporal coincidence.

Response: We thank the reviewer for pointing out this inaccurate statement. In the revised manuscript, we have deleted this sentence.

5. Line 230: what does "high-stain-rate regions" mean? Please explain.

Response: We thank the reviewer for noticing this typo. The correct statement is "high-strain-rate regions", which means that the regions are subjected to high strain rate (or high rate of deformation of wind flows). We have now corrected this error as "high-strain-rate regions" in the revised manuscript. Please see line 323.

6. Line 246: consider deleting the word "easily". I believe that it is a substantial effort.

Response: Deleted as suggested.

7. Lines 218, 220: Recommend adding references to recent works on dust storms, and addressing their findings in the context of the discussion:

Silva, H.G., Lopes, F.M., Pererira, S., Nicoll, K., Barbosa, S.M., Conceicao, R., Neves, S., Harrison, R.G., Pereira, M.C. (2016). Saharan dust electrification perceived by a triangle of atmospheric electricity stations in Southern Portugal. *J. Electro.* 84, 106–120.

Katz, S. Y. Yair, C. Price, R. Yaniv, I. Silber, B. Lynn and B. Ziv (2018), Electrical properties of the 8-12th September, 2015 massive dust outbreak over the Levant. *Atmos. Res.*, 201, 218-225.

Solomos, S., Ansmann, A., Mamouri, R.-E., Biniotogulou, I., Patlakas, P., Marinou, E., Amiridis, V., (2017). Remote sensing and modeling analysis of the extreme dust storm hitting the Middle east and eastern Mediterranean in September 2015. *Atmos. Chem. Phys.* 17, 4063–4079.

Yair, Y., S. Katz, R. Yaniv, B. Ziv and C. Price (2016), An electrified dust storm over the Negev desert, Israel. *Atmos. Res.*, 181, 6-71

Response: We thank the reviewer for pointing out these very important studies that are highly relevant to our work. We have discussed and cited these studies in the revised manuscript. Please see Refs. 21-24 in the revised manuscript.

Reviewers' Comments:

Reviewer #2:

Remarks to the Author:

The authors responded accurately and convincingly to all the doubts I posed in the previous revision of the manuscript.

The manuscript results now significantly improved and in my opinion it is suitable for publication.

Reviewer #3:

Remarks to the Author:

Let me congratulate the authors for an excellent work.

The revised manuscript is much improved and the responses are adequate, detailed and complete. I have no additional comments to offer, and would therefore recommend that the paper is accepted for publication in its present form.

We thank the reviewers for their kind work and positive feedback. The following text contains the reviewers' comments (in black) and our responses (in blue).

Reviewer #2 (Remarks to the Author):

1. The authors responded accurately and convincingly to all the doubts I posed in the previous revision of the manuscript.

Response: We are very pleased that our responses and revisions can be fully accepted by the reviewer. We thank the reviewer for her/his constructive comments in the previous peer review stage.

2. The manuscript results now significantly improved and in my opinion it is suitable for publication.

Response: We thank the reviewer for the kind recommendation of our work.

Reviewer #3 (Remarks to the Author):

1. Let me congratulate the authors for an excellent work.

Response: We thank the reviewer for her/his positive appreciation of our work.

2. The revised manuscript is much improved and the responses are adequate, detailed and complete. I have no additional comments to offer, and would therefore recommend that the paper is accepted for publication in its present form.

Response: We are very pleased that our responses and revisions can be fully accepted by the reviewer. We thank the reviewer for her/his constructive comments in the previous peer review stage. We also thank the reviewer for the kind recommendation of our work.